

# Mid-Holocene climate of the Tibetan Plateau and hydroclimate in three major river basins based on high-resolution regional climate simulations

Yiling Huo[1], William Richard Peltier[1], and Deepak Chandan[1]

[1] Department of Physics, University of Toronto, Toronto, M5S 1A7, Canada

*Correspondence to*: Yiling Huo (yhuo@physics.utoronto.ca)

**Abstract.** The Tibetan Plateau (TP) exerts strong influence on both regional and global climate through thermal and mechanical forcings. The TP also contains the headwaters of large Asian rivers that sustain billions of people and numerous
ecosystems. Understanding the characteristics and changes to the hydrological regimes on the TP during the mid-Holocene (MH) will help understand the expected future changes. Here, an analysis of the hydroclimates over the headwater regions of three major rivers originating in the TP, namely the Yellow, Yangtze and Brahmaputra rivers is presented, using an ensemble of climate simulations, which have been dynamically downscaled to 10-km resolution with the Weather Research and Forecasting Model (WRF) coupled to the hydrological model WRF-Hydro. Basin-integrated changes in the seasonal
cycle of hydroclimatic variables are considered. In the global model, we have also incorporated Green Sahara (GS) boundary conditions in order to compare with standard MH simulations (which do not include GS) and to capture interactions between the GS and the river hydrographs over the TP. Model-data comparisons show that the dynamically downscaled simulations significantly improve the regional climate simulations over the TP in both the modern day and the MH, highlighting the crucial role of downscaling in both present-day and past climates, although both global and regional models have a cold bias
in modern-day simulations and underestimate the wet anomalies inferred from proxy data in the east and southeast part of the TP. TP precipitation is also greatly influenced by the inclusion of a GS, with a particularly large increase predicted over the southern TP, as well as a delay in the monsoon withdrawal. The model performance was first evaluated over the upper basins of the three rivers before the hydrological responses to the MH forcing in streamflow as well as temperature, rainfall and snowmelt for the three basins were quantified via the WRF simulations.

# 1 Introduction

The Tibetan Plateau (TP) and its surrounding regions, often referred to as the Third Pole, with an average elevation exceeding 4000 m and an area of $2.5 \times 10^6$ km², constitute the most extensive and highest plateau in the world. The TP also has the most extensive cryosphere outside the Arctic/Greenland and the Antarctic regions (Yao et al., 2012), including snow





cover, glaciers and permafrost. As a unique geological and geographical unit, the TP exerts a profound influence on the
Asian and global climate through mechanical and thermal forcing mechanisms (Kutzbach et al., 1993; An et al., 2001;
Molnar et al., 2010). Climate over the TP, which is characterized by a wet and warm summer and a cool and dry winter, is
extremely sensitive to global climate change and has experienced rapid and pronounced changes in recent decades (Lehnert
et al., 2016; Yao et al., 2019). With an elevation gradually decreasing from the northwest to the southeast, the TP also acts as
the "water tower" and the source region of major Asian rivers that flow down to almost half of humanity, including the
Yellow, Yangtze and Brahmaputra (Yarlung Tsangpo) Rivers (Immerzeel et al., 2020).

The Yellow River, which is often referred to as the cradle of the Chinese civilization, originates in the northern TP (Fig. 1)
and supports 30% of the China's population (Huang et al., 2009). It passes through the Loess Plateau and the North China
Plain, before emptying into the Bohai Sea on the east coast of China. The Yellow River's headwaters are situated in the
Bayan Har Mountains in the northern TP. Its water resources are vital to the fragile and unique temperate, alpine, and
wetland ecosystems within the upper Yellow River Basin (UYEB) and are extremely sensitive to climate change. The
longest river in Asia, the Yangtze, rises in the Tanggula Mountains over the eastern part of the TP and flows eastwards to the
East China Sea (Fig. 1). Its drainage basin comprises one-fifth of China's land area, which is home to nearly a third of the
country's population. The Brahmaputra River, which originates in the southern TP and flows eastwards before turning
southward into India in Arunachal Pradesh, is an important trans-boundary river. After entering India, it turns southwestward
and eventually enters Bangladesh where it merges with the Padama River in the Ganges Delta. Dramatic climatic
heterogeneity exists within its drainage area, from the dry and cold upper region with an elevation higher than 5000 m,
which features glaciers and permanent snow, to the humid and hot downstream area that is only 200 m above sea level. The
Brahmaputra River flows across four Asian countries including countries with the two largest populations in the world,
China and India. Understanding hydrological processes of these rivers is necessary for informed current and future
sustainable management of their water resources and are of significant economic importance for these countries. Thus, it is
important to understand how climate change will impact streamflow and, hence, available water resources on a basin scale.

Meteorological observations on the TP indicate a surface warming trend that is twice as large as that observed globally
(IPCC, 2013), accompanied by spatial differences in changes to precipitation and glaciers (Yao et al., 2012). An increasing
trend in precipitation has been observed to be occurring in the western and central TP and a smaller precipitation increase
has been detected in the eastern TP (Zhou et al., 2019). Most regions of the TP have become warmer and wetter during the
past three decades, while the southern TP has become warmer and drier (Yang et al., 2014). Glacier melt is an important
hydrologic process in cold regions, that significantly modifies streamflow characteristics, such as quantity, timing, and
variability of flows (Kaser et al., 2010; Immerzeel et al., 2020). Changes in temperature and precipitation are expected to
severely affect the melting characteristics of mountain glaciers (Barnett et al., 2005; Immerzeel et al., 2020), inducing
heterogenous responses over different parts of the TP. In a warmer climate, glacier melt will be enhanced due to the
disappearance of fresh snow and the associated decrease of the surface albedo (Barnett et al., 2005). Glacier melt particularly
affects streamflow in warm and dry seasons (Kaser et al., 2010) and will increase the downstream discharge of the rivers



(Yao et al., 2004). However, this increase is transitory and is expected to cease when the glaciers have largely retreated (Yao and Yao, 2010). Over a long time period, mountain glaciers are predicted to disappear, leading to a decline in river flow in

the summer season (Immerzeel et al., 2020) and posing serious and unprecedented threat to access to water resources in the TP and downstream regions (Kehrwald et al., 2008).

Rivers that originate in the TP often suffer from a scarcity of data required for hydrological simulation and water resource assessment due to the extreme conditions under which measurements must be acquired. Regional hydrological modeling in these ungauged regions has therefore attracted increased attention in water resource management research but faces

significant technical challenges due to the complex regional topography. The resolution of current global climate models (GCMs) is not sufficient for this area, because regional climate over the TP is strongly affected by the steep topographic gradients that are poorly resolved in global models. This hinders our ability to characterize the water cycle, study climate change impacts in this area and find solutions to international water-resource issues. For this reason, global climate simulations need to be downscaled to scales at which important topographic features are better resolved.

The study of past climate conditions is an important tool in any effort to understand the mechanisms influencing present and future climate and its variability. In addition, understanding the range of variability will help to improve climate models that are used to simulate the regional response to a variety of abrupt and long-term changes in temperature, atmospheric circulation, and moisture sources, which in turn regulate the river discharge flowing out of the TP. The mid-Holocene (MH, approximately 6000 years ago) is one of the most widely studied periods of the Quaternary and is a period of focus for the

Paleoclimate Modelling Intercomparison Project (PMIP). During the MH variation in the Earth's orbital configuration led to differences in the latitudinal and seasonal distribution of incoming solar radiation (Berger 1978), resulting in a climate that was remarkably different from that of the present day. During summer (winter), the insolation was anomalously strong (weak) over the Northern Hemisphere (NH) leading to an enhancement of the seasonal cycle. A temperature optimum for the TP during the MH has been recorded by pollen data (Herzschuh et al., 2006; Lu et al., 2011; Ma et al., 2014). MH climatic

changes over the TP are crucial to understanding the impact of the recent/modern global warming and to predicting future climate change (Wanner et al., 2008). During the MH, South and East Asian monsoons are expected to have been enhanced due to the increased boreal summer insolation (Huo et al., 2021; Herzschuh, 2006; Maher, 2008). However, compared with the reconstructions, regional biases and large spreads are found in numerical simulations of the MH climate by the PMIP models, in particular, over and around the TP (Zheng et al., 2013).

During the MH, changes in insolation and albedo feedbacks also strengthened the African monsoon, making northern Africa much wetter than today (Skinner and Poulsen, 2016). Lakes, rivers, and wetlands formed and developed in the now arid regions of northern Africa and led to a northward extension of the vegetation cover over the Sahara, i.e., a "Green Sahara" (GS; Pausata et al., 2020; Chandan and Peltier, 2020). Studies have shown that the remote feedbacks associated with a vegetated Sahara and the strengthened African monsoon can alter the Walker circulation through changes in tropical Atlantic

SSTs, which in turn enhanced the Asian monsoon (Pausata et al., 2017). The TP, which is situated within the region influenced by the Asian monsoon (Conroy and Overpeck, 2011) is thus also affected by the greening of the Sahara. One of




the key goals of the present study is to outline the remote response of the MH TP climate to the GS teleconnection by analysing a set of simulations in which the land cover is changed from desert to shrubland over a large part of northern Africa (Chandan and Peltier, 2020) and. Since few studies have investigated the runoff characteristics over the TP under the
MH climate conditions, in this study we also analyse the hydrological response of the TP rivers to the MH climate forcing. The objective of this article is thus to first evaluate the performance of the coupled UofT-CCSM4-WRF-Hydro model over the TP and then analyse the response of hydroclimate over the three rivers over the TP to MH orbital forcing and remote GS feedback, based on high-resolution, dynamically downscaled regional climate simulations. Details concerning the experimental design and the model configurations used in the downscaled simulations are given in the next section (Sect. 2).
The validation of the results is discussed in Sect. 3, before the general patterns of MH anomalies are outlined. The central point of this study is hydroclimatological anomalies in the UYEB, UYAB and UBB, which is presented in Sect. 4. Section 5 summarizes the main results of the present study.

## 2 Model descriptions and experimental design

### 2.1 Experimental design

This study employs dynamical downscaling, a modelling methodology based upon the use of a dynamically consistent, physically based high-resolution regional climate model (RCM) to downscale global climate simulations so as to better account for regional feedbacks in the climate system. The global climate simulations used to drive the RCM have been obtained using the University of Toronto version of the National Center for Atmospheric Research (NCAR) Community Climate System Model version 4 (UofT-CCSM4; Peltier and Vettoretti, 2014; Chandan and Peltier, 2017, 2018). The global
climate simulations are then dynamically downscaled to 10-km resolution using the Weather Research and Forecasting (WRF) model version 4.1 with four different convection parameterization schemes, constituting a small physics ensemble. We have also coupled WRF-Hydro (Gochis et al., 2020) to the WRF model as a hydrological extension package for hydrometeorological (e.g., rainfall, runoff, groundwater flow, streamflow) simulations, which will also be a primary focus of the analyses to be discussed herein. The fully coupled WRF-Hydro modelling system outperforms the WRF-only model and
thus has been employed in atmosphere-hydrology simulations over various domains around the globe (Senatore et al., 2015; Kerandi et al., 2017; Somos-Valenzuela and Palmer, 2018). We focus our hydroclimatic analysis on the upper Yellow, Yangtze and Brahmaputra River basins (UYEB, UYAB and YBB, respectively), the headwaters of which are all located on the TP.

Four global climate simulations were performed using UofT-CCSM4 at approximately 1° resolution, one for the historical
period (1980-94), one for the preindustrial (PI) reference period and two for the MH. UofT-CCSM4 is based on the standard CCSM4 (Gent et al., 2011), but specific modifications have been made to the ocean component to make it more appropriate for paleoclimate simulations (Peltier and Vettoretti, 2014; Chandan and Peltier, 2017). The same PI and MH global simulations have also been employed in Huo et al. (2021) for the purpose of studying MH monsoons in South and Southeast





Asia. The two MH simulations differ from each other in terms of the prescribed land surface modifications over northern
Africa (Chandan and Peltier, 2020) but otherwise share the same orbital parameters and trace gases as the PMIP4
recommendations (Otto-Bliesner et al., 2017): $MH_{REF}$ is the "control MH" which uses the same land surface as in PI, while
$MH_{GS}$ extends $MH_{REF}$ by adding the GS vegetation.

Each UofT-CCSM4 simulation was separately downscaled over the TP for 15 years, using the WRF model with a nested
configuration consisting of an outer and an inner domain. The outer domain covers most of Asia at 30-km resolution, while
the inner domain encompasses the TP, as well as parts of the surrounding territory, at 10-km resolution. The outlines of the
two domains as well as the topography in the WRF outer domain are shown in Fig. 1. The analysis presented in this article is
primarily based on a four-member physics ensemble of dynamically downscaled climate simulations, with the same choice
of major physics parameterizations as in Huo et al. (2021), where the MH climate anomalies over the South and Southeast
Asia were studied. The simulations in each regional climate ensemble employ different cumulus parameterizations in WRF
as in Huo et al. (2021), namely the Kain-Fritsch scheme (KF; Kain, 2004), the Grell–Freitas ensemble scheme (GF; Grell
and Freitas, 2014), the Tiedtke scheme (Tiedtke, 1989) and the Betts–Miller–Janjić scheme (BMJ; Janjić, 1994), making a
mini-physics ensemble for the TP, that allows us to study the sensitivity of model performance to different cumulus
parameterizations and thereby to estimate the uncertainty associated with these parameterizations on the simulated MH
climate.

The dynamical downscaling methodology employed here is a somewhat further developed version of the dynamical
downscaling "pipeline" originally introduced in Gula and Peltier (2012) and then applied in d'Orgeville et al. (2014), Erler
and Peltier (2016), Peltier et al. (2018), Huo and Peltier (2019, 2020, 2021), where the expected climate change impacts over
the Great Lakes Basin of North America, over western Canada, upstream and downstream of the Rocky Mountains
topographic barrier and over the South and Southeast Asia were investigated. In the present version of this pipeline, WRF-
Hydro V5.1.1 (Gochis et al., 2020) is coupled to WRF for the purpose of performing hydrometeorological simulations. The
WRF-Hydro model is a multi-scale and multi-physics hydrologic model community model widely used for flash flood
prediction, seasonal simulation of water cycle components and regional hydroclimate impacts assessment both in the short
term and long term. In this study, the Noah Land Surface Model (Tewari et al., 2004) is used in the coupled WRF-Hydro
modelling system and high-resolution hydrological routing modules have been adopted to represent subsurface lateral flow
(Gochis and Chen, 2003). Spin-up simulations of 5 years have also been conducted to achieve numerical stability of the
model outputs.

## 2.2 Observational datasets

To validate the simulations of historical temperature and precipitation against observations, the CRU dataset V4.05 (Harris et
al., 2020) at 0.5° resolution is employed. This widely used climate dataset was derived by the interpolation of monthly
climate anomalies from several extensive networks of weather stations. To compute differences between model results and
the observations, the latter have been reprojected and resampled onto the native grid of each model.



In addition, to evaluate the simulated hydrological metrics, mean monthly streamflow data from three stations have been obtained from the Global Runoff Data Centre (GRDC, 2015). These gauging stations (Tanglai Qu, Zhimenda and Yangcun) are located at the basin outlets of the UYEB, UYAB and UBB (Fig. 1) respectively and observations are available from 1978 to 1997, from 1978 to 1997 and from 1956 to 1982, respectively. Not all years in the station record have complete monthly data; nevertheless, all available flow data have been used to compute the average monthly discharges discussed in Sect. 4.

## 3 Historical and MH climates

This section presents an overview of the validation of our methodology against historical observations as well as results from the MH simulations. Note that all temperature and precipitation biases and anomalies reported in this section refer to spatial averages over the inner WRF domain.

### 3.1 Validation against historical observations

Figure 2 shows the JJAS (June-July-August-September) and DJF (December-January-February) temperature biases, compared to the CRU dataset, in the of UofT-CCSM4 historical simulation and in the WRF inner-domain ensemble average for that same time period. The temperature biases are evidently dependent on the season. In JJAS the bias pattern is fairly consistent between the global and the regional models and includes a cold bias over the high elevations of the TP and a warm bias in the northeast of the TP and the lower lands around the TP, especially in the Tarim Basin. Overall, summer temperatures are simulated fairly well by both models and integrated over the entire inner WRF domain, the average temperature bias in both models is less than 0.5∘C. However, during winter, the GCM features a strong cold bias over the TP (> 3∘C averaged over the inner domain), whereas the regional model has a better representation of DJF temperature with a significantly smaller average bias of approximately 1∘C. Since most of the precipitation over the TP occurs in summer, Fig. 3 shows the distribution of the JJAS average total precipitation in the observational dataset and the biases in the global model and the WRF ensemble. The spatial patterns in both models and observations are strongly dominated by orographic forcing with very high precipitation intensities on the south side of the Himalayas in contrast with the very low precipitation rates in the rain shadows of the mountain ranges, in particular over the northern TP which constitutes a major part of the UYEB and UYAB. There is moderate rainfall over the Sichuan Basin east of the TP, and the southern plateau, where the UBB is located. However, the representation of orographic precipitation is strongly dependent on model resolution: the low-resolution global model significantly underestimates both the peak intensities in front of the Himalayas as well as the rain shadow effect in its lee. The representation of orographic impact on precipitation is dramatically improved in WRF at 10-km resolution. The wet bias over the northern and eastern TP has also been eliminated in the RCM. Averaged over the inner WRF domain, the JJAS precipitation bias is 0.4 mm d$^{-1}$ in the WRF ensemble average and 1.4 mm d$^{-1}$ in the driving UofT-CCSM4 simulation. Both models show a similar wet bias in winter (Fig. 3d), and such excessive snow possibly contributes to the lower winter temperature.





## 3.2 Simulated MH climate

We next turn to an attempt to analyse the climate anomalies in MH$_{REF}$ and MH$_{GS}$ and to investigate the impact of the
Saharan vegetation on the TP.

The simulated JJAS temperature over the TP in our MH ensembles is shown in Fig. 4. Due to the change in the Earth's orbital parameters, during the MH the NH receives more insolation during boreal summer that leads to a significant warming over the TP in both the GCM and the WRF ensemble (Figs. 4a and 4b). The temperature response also shows cooling south of the TP and over the ocean (Huo et al., 2021). Thus, the land-sea thermal contrast is increased in the MH favouring the
enhancement of the monsoon circulations. The winter temperature over the TP shows a general cooling over the TP (Fig. 4e) that is associated with the decrease in winter insolation in the NH. These changes imply that the solar radiation is one of the most important factors dominating the temperature changes in climate models. Comparing the JJAS temperature changes produced by the GCM and RCM, the WRF ensemble shows an overall stronger warming over the TP, particularly in the northern part. Moreover, the regional model also produces a larger warm anomaly than the global model from mid-summer
to early-winter (Fig. 4e). Taking the influence of a vegetated Sahara into account results in higher temperature over the TP all year round, except in winter when the difference is negligible (Figs. 4d and 4e). The influence of MH$_{GS}$ in both models is characterized by an overall increase in the annual mean temperature over the TP, which agrees with estimates from pollen records (Shi et al., 1993), while an annual cooling is obtained with MH$_{REF}$. Averaged over the WRF inner domain, the JJAS temperature anomalies of the WRF ensemble mean are 1.1 ∘C with MH$_{REF}$ and 1.8 ∘C with MH$_{GS}$. Particularly, over the
norther plateau, the warming is significantly enhanced from approximately 2 ∘C to 4 ∘C, leading to a better agreement with proxy-based estimation of 4–5 ∘C warming (Shi et al., 1993) over the TP.

MH orbital and GHG forcings result in wet anomalies in both global and regional models (Figs. 5a and 5b). The enhanced annual precipitation in the southern parts of the TP in MH$_{REF}$ broadly agrees with the proxy reconstructions by Bartlein et al. (2011) and Herzschuh et al. (2019). However, with orbital forcing alone (MH$_{REF}$) both models fail to reproduce the wet
anomaly in the east and southeast parts of the TP, although the area with reduced rainfall in the downscaled results is smaller than that in UofT-CCSM4 and thus agrees better with the overall wet signal suggested by reconstructions. Considering all the points with proxy data from Bartlein et al. (2011) within the WRF inner domain, the WRF ensemble average has a bias of 0.6 mm d$^{-1}$, which is slightly smaller than that of the global model (0.7 mm d$^{-1}$). Under GS forcing (MH$_{GS}$), precipitation is simulated to increase over almost the entire domain except a small patch in the southeast (Fig. 5d). The inclusion of GS
surface boundary conditions greatly expands and intensifies the wet anomalies in WRF, further reducing the bias with respect to the reconstruction by Bartlein et al. (2011) by half. Reconstructions (Shi and Song, 2003; Sun et al., 2006) suggested about 20–50% higher annual precipitation over Daihai Lake (113∘ E, 41∘ N) and Diaojiao Lake (112∘ E, 41∘ N) during the MH. WRF produced precipitation increases of 38% and 30% at these two locations when the influence of a vegetated Sahara is considered, which agrees with the range of reconstructed precipitation changes. However, the
precipitation increase rates at these two locations are significantly underestimated by both UofT-CCSM4 in MH$_{GS}$ (Fig. 5c;





less than 20%) and the WRF simulation without GS forcing (Fig. 5b; less than 5%). Comparison of downscaled results including the GS with paleoclimate reconstructions shows significant improvements, especially over the north and west sides of TP. These results highlight the climate sensitivity to Saharan vegetation changes via ocean-atmosphere teleconnections and emphasized the importance of incorporating MH vegetation feedbacks.

The spatial patterns of JJAS precipitation anomalies resemble those of the annual mean but with a larger magnitude (Fig. 6), which is expected since annual precipitation largely results from the summer rainfall over the TP, a region heavily influenced by the Asian monsoon circulation. The seasonal cycle of precipitation obtained with $MH_{REF}$ changes in response to the insolation changes at 6 ka (Fig. 6c): rainfall slightly decreases before May and then strengthens with the onset of the monsoon. Meanwhile, the retreat of monsoon is also delayed by about one month as the increased rainfall persists until

October. Such changes are consistent with the results obtained in PMIP models (Zheng et al., 2013). Inclusion of the GS boundary conditions not only leads to a weaker decrease in rainfall than $MH_{REF}$ in spring, but also eliminates the negative rainfall anomaly in May in $MH_{REF}$. During the monsoon season (JJAS), the precipitation increases in $MH_{GS}$ are more than two times as large as those caused by the MH orbital and GHG forcings alone in both the UofT-CCSM4 and WRF physics ensemble members over the TP. Averaged over the WRF inner domain, JJAS precipitation is generally found to increase by

0.5 mm $d^{-1}$ (14%) in $MH_{REF}$ and 1.1 mm $d^{-1}$ (31%) in $MH_{GS}$ in the WRF ensemble average. Comparing different WRF ensemble members, they all share similar general characteristics, but the magnitude of the absolute precipitation changes depends on the individual ensemble member (Figs. 6c and 6f). The fourth ensemble member using the BMJ cumulus scheme produces the smallest absolute JJAS precipitation increases in both $MH_{REF}$ (0.35 mm $d^{-1}$) and $MH_{GS}$ (0.87 mm $d^{-1}$) experiments, while the KF scheme in the first ensemble member and the GF scheme in the second ensemble member

produce the largest rainfall increase in $MH_{REF}$ (0.59 mm $d^{-1}$) and $MH_{GS}$ (1.31 mm $d^{-1}$), respectively. The Sahara greening also intensifies the precipitation at the end of the monsoon season (September and October) over the TP, further postponing the monsoon withdrawal, which is similar to the findings in our previous studies over South and Southeast Asia (Huo et al., 2021).

## 4 Three river basins

In this section, the simulated climate for the present day as well as the MH over the three river basins on the TP are discussed. The hydroclimatic analysis presented here are based on surface fluxes and climatic state variables averaged over each river basin. Figures 7, 9 and 11 show the seasonal cycles of hydroclimatic variables over the UYEB, UYAB and UBB during the historical period and Figs. 8, 10 and 12 show the anomalies during the MH. In these figures, the WRF ensemble averages are plotted in thick black lines while results from individual ensemble members are plotted in colors (see figure

legends). All variables were averaged over the entire basin area and over each 15-year simulation period. Observations are shown by open circles. Only temperature, total precipitation, and river discharge are available from observations. The observed temperature and precipitation values have been assembled from gridded CRU observational datasets and averaged





over the area of the basins. The river discharges, recorded by gauging stations, have been normalized by the associated basin area. Areas of contribution for UYEB, UYAB and UBB are approximately 286000 km$^2$, 138000 km$^2$ and 153000 km$^2$, respectively. The error bars on the observational data are based on the standard error of the mean (SEM), which is defined as $\sigma/\sqrt{n}$, where $\sigma$ is the sample standard deviation and $n$ is the sample size.

Solid precipitation in panel b of Figs. 7-12 is snow and graupel, and net precipitation (panel c in Figs. 7-12) is defined as total precipitation minus evapotranspiration. Note that evapotranspiration, snowfall, and snowmelt are calculated by the Noah land surface model, while runoff is computed by the WRF-Hydro model.

For comparison between the preindustrial and MH simulations, we applied the linear scaling method to the original streamflow results. Due to the large (negative) JJAS runoff bias in WRF during the historical period, especially in the UYEB and UYAB (Figs. 7d and 9d), it was deemed necessary to apply a simple form of bias correction to rescale the monthly river discharge so that the monthly watershed average for the historical period matches the observations (using one factor per month). Linear scaling corrects the ensemble mean values of the historical simulations based on monthly errors (Trambauer et al., 2015). The scaling factor was first obtained through calculating the ratio between the observed and modelled historical values. Then the monthly scaling factor was applied to each uncorrected daily runoff simulations of that month so that the monthly mean values of the historical simulations match those of the observation. The streamflow values of the MH simulations have been bias corrected using the same factors as for the historical results so that they can be usefully compared.

## 4.1 The UYEB

The Yellow River originates in the northern TP and runs eastward across the Loess Plateau and the North China Plain, eventually draining into the Pacific Ocean at the Bohai Gulf on the east coast of China. The UYEB above the Tanglai Qu hydrometric station, with an altitude varying from 4.9 km in the southwest to 1.1 km in the northeast, is situated on the northeast side of the TP but also covers a small section of the western Loess Plateau (Fig. 1). It is situated within a semiarid region with an annual precipitation of approximately 400 mm and an annual mean air temperature of approximately 2 °C based on the CRU dataset.

The seasonal cycles of hydroclimatic variables over the UYEB are provided in Fig. 7. With the exception of winter, the seasonal cycle of temperature (Fig. 7a) is very accurately reproduced by the WRF ensemble. The GCM temperature is lower than the observation by 3.8 °C in DJF but the WRF model is able to reduce this cold bias to approximately 1.4 °C. The large cold bias in UofT-CCSM4 is likely due to excess winter precipitation (Fig. 7b) combined with delayed snowmelt, which results in a snow–albedo feedback on temperature. The JJAS temperatures simulated by both the global and the regional models fit the observation well. The Asian monsoon exerts strong influence on the UYEB: about 70% of annual precipitation falls in JJAS and less than 2% occurs in the winter months (DJF) according to the observation. The simulated seasonal cycle of precipitation in WRF is too strong compared to the observation, owing largely to excessive precipitation in summer (the wettest season). This bias is inherited from the GCM and although downscaling greatly reduces the bias, precipitation is still overestimated for JJAS over the UYEB (Fig. 7b). There is, as expected, little solid precipitation in summer and all





precipitation falls as solid precipitation during winter, which leads to a snowmelt-driven runoff peak in late spring and early summer (Fig. 7c). It is interesting to note that the spread among the ensemble members is significantly larger for liquid precipitation than for solid precipitation or temperature; different convection schemes greatly affect the JJAS precipitation and the second ensemble member using the GFE cumulus scheme exhibits the largest wet bias. Net precipitation pattern is

characterized by two peaks in June and September due to the large evapotranspiration in July and August (Fig. 7c). The seasonal streamflow for the UYEB during the historical period exhibits a broad shape with flow increasing slowly in April and slowly declining after October. Runoff in spring clearly follows the timing of snowmelt. Streamflow regime is also characterized by double peaks, in July and September, respectively (Fig. 7d). The good correspondence between the streamflow and net precipitation regimes in summer indicates that the monsoon rainfall is the dominant contributor to runoff

over the UYEB. WRF-Hydro is able to reproduce runoff that broadly agrees with observed river discharge. However, the model tends to underestimate the magnitude of the runoff peaks.

During the MH, the overall changes in mean temperature in the UYEB follow changes of insolation the magnitude of which increases during boreal summer and decreases in winter. The WRF model produces larger positive temperature anomalies in late summer and smaller cold anomalies in fall over the UYEB compared to UofT-CCSM4 (Fig. 8a). Compared with $MH_{REF}$,

temperature changes in experiments with the GS forcing have very similar characteristics, but the simulated temperature is generally warmer than in $MH_{REF}$ (Figs. 8a and 8b). Warming also starts earlier in May instead of June and ends one month later in November and the peak warming also occurs one month later in our WRF $MH_{GS}$ experiments relative to the $MH_{REF}$. Both the sign and magnitude of the temperature changes are similar between the WRF simulations based on different physics configurations in both MH ensembles.

The changes in monthly precipitation over UYEB are shown in Figs. 8c and 8d. The $MH_{REF}$ simulations in both UofT-CCSM4 and WRF show a significant decrease in precipitation except in September and October. The absolute regional dry anomaly is the largest in August, while September experiences the largest increase in rainfall. When vegetation is imposed over the Sahara, the negative peak in August and the positive peak in September still exist, but there is another smaller precipitation increase peak in June, showing a strong rainfall enhancement in late spring and early monsoon season in

response to the GS forcing. However, averaged over the whole monsoon season, the WRF ensemble mean in $MH_{GS}$ shows no significant precipitation change ($< 3\%$) due to the offset between these positive and negative anomalies, while the WRF simulations that only account for the insolation forcing ($MH_{REF}$) produce a decrease in monsoonal precipitation (9%). Another major change in the hydrological cycle of the UYEB is the shift of part of the solid precipitation to liquid form in spring and summer in $MH_{GS}$ when compared to $MH_{REF}$, which in turn will lead to a decline in snowmelt (Figs. 8e and 8f).

The reduction in peak snowmelt due to the combined effects of MH insolation forcing and GS forcing is close to 50%. At the same time higher temperatures during the warm season resulting from the MH insolation forcing also lead to larger evaporation, which accounts for the drop in net precipitation and also the decrease of the UYEB discharge especially in summer (Figs. 8g and 8h). The magnitude of streamflow reductions in both MH experiments are similar, a consequence of the larger temperature rise in $MH_{GS}$ and the opposite and compensating effects of larger snowmelt and greater evaporation,





though the peak of streamflow decrease in MH$_{GS}$ occurs one month later compared to the MH$_{REF}$ in late monsoon season
       (September). Such reduction in the upper Yellow River runoff has also been found over the past decades, which is related to
       rising temperatures and decreasing wet season precipitation (Cuo et al., 2013).

## 4.2 The UYAB

       The UYAB lies in the eastern part of the TP and is influenced by the Asian monsoon (Fig. 1). The basin, whose altitude
varies from 5.7 km in the west to 4.2 km in the east, drains past the Zhimenda hydrological station (Fig. 1). The integrated
       precipitation in JJAS accounts for more than 85% of the total annual precipitation (Fig. 9b) and the JJAS discharge accounts
       for more than 70% of the annual total at Zhimenda (Fig. 9d).

       Figure 9 shows the seasonal cycles of basin temperature, precipitation, snowmelt and streamflow in the UYAB. It is evident
       that the seasonal cycle in temperature is quite well reproduced, except for a larger cold bias in winter than the UYEB, even
though the downscaled simulations have already halved the DJF cold bias in the global simulation. It is interesting to note
       that the WRF ensemble actually suffers from a stronger cold bias than the UofT-CCSM4 in spring (−3 ◦C in March-April-
       May). Note here, UYAB covers a relatively smaller area, compared to other basins, and the station density in the UYAB is
       relatively low (Harris et al., 2020), so that the observational error can be large. Like the UYEB, precipitation in the UYAB
       peaks in summer and is relatively low in winter (Fig. 9b); qualitatively the seasonal cycle is well captured in WRF, but the
contrast between summer and winter is still overestimated in all WRF ensemble members. In all WRF ensemble members,
       the seasonal precipitation over the UYAB is somewhat better represented in the third WRF ensemble member with the
       Tiedtke cumulus scheme whereas the GFE scheme used in the second ensemble member produces the largest excess in JJAS
       precipitation. UofT-CCSM4, however, is characterized by a much greater wet bias, with significantly more precipitation in
       spring and summer. The improvement in simulation of both regional temperature and precipitation indicates the fidelity
brought by the use of higher resolution. From December to April, all precipitation occurs as snow. In summer, precipitation
       primarily falls as liquid rain, but there is still snow even in July and August because of the colder climate over the UYAB
       compared to the UYEB. Snowmelt over the UYAB is again characterized by double peaks in June and September (Fig. 9c),
       which correspond to the two peaks in snowfall (Fig. 9b). Therefore, the melting of fresh snowfall is likely the major
       contributor to the snowmelt peaks simulated by the model. Even though the simulated precipitation maximum in the UYAB
occurs in August, the net precipitation peaks in September due to the great evapotranspiration in August (Fig. 9c). The
       observed streamflow in the UYAB rapidly rises in June and reaches a maximum in July, and flow level remains relatively
       high until September (Fig. 9d). The monthly streamflow during the historical period was reasonably simulated by the WRF-
       Hydro model, but there is significant underestimation of streamflow in July and August (Fig. 9d). This is likely a result of
       the stronger evapotranspiration, leading to a lower runoff. The mismatch in runoff peak magnitude compared to the
observation is similar in all WRF configurations except the second ensemble member using the GFE cumulus scheme; the
       GFE cumulus scheme overestimates the runoff all year round and also produces the largest net precipitation and snowmelt
       among all four WRF ensemble members.





The changes to the surface air temperatures in UYAB during the MH are also determined by the changes to the pattern of insolation (Figs. 10a and 10b). In both $MH_{REF}$ and $MH_{GS}$ experiments, the temperature anomalies produced in WRF are

larger than the UofT-CCSM4 all year round, especially in fall. During the MH, temperature is simulated to have increased by 1.6 °C in the monsoon season in the WRF ensemble average in $MH_{REF}$ (compared to 1.0 °C in UofT-CCSM4), and 3.1 °C in $MH_{GS}$ (compared to 2.2 °C in UofT-CCSM4). In comparison, the ensemble mean for the MH experiments which assumes the same Saharan vegetation cover as the PI experiences a small annual cooling over the UYEB (−0.2 °C), while including a green Sahara flips the annual temperature anomaly into a warming of 1.3 °C. JJAS precipitation changes in the WRF

ensemble mean are −9.8% in $MH_{REF}$ and 5.3% in $MH_{GS}$ (Figs. 10c and 10d), and such a transition from a dry to a wet anomaly is clearly associated with the strengthening of the Asian monsoon resulting from the greening of the Sahara (Pausata et al., 2017; Huo et al., 2021). UofT-CCSM4, on the other hand, simulates almost no change in summer precipitation for the UYAB in $MH_{REF}$ and a much larger increase in the MH experiment with a vegetated Sahara. However, considering the much better representation of summer precipitation in WRF compared to UofT-CCSM4 (Figs. 3 and 9), we

suggest that this may actually be an artifact of the coarse resolution employed in the GCM. Note, however, that because of the relatively smaller size of the basin, total precipitation changes over UYAB have a larger spread between the WRF ensemble members than in UYEB, but the temperature-induced change from solid to liquid precipitation and particularly the snowmelt decrease for most months are very robust among all WRF ensemble members (Figs. 10e and 10f). The reduction in annual snowmelt in the WRF ensemble is approximately 17% in $MH_{REF}$ and 19% in $MH_{GS}$. These changes combined with

the generally increased evaporative water losses cause a substantial decrease in runoff, especially in summer (Figs. 10g and 10h). Such negative anomaly in JJAS discharge agrees with CMIP5-based projections (Krysanova et al., 2017) and hydrological model results forced by PMIP4 data during the last interglacial (LIG; Scussolini et al., 2020). Compared with UYEB, the reduction in snowmelt plays a more important role in the runoff decrease over the UYAB. The magnitudes of UYAB runoff changes in both MH simulations are similar because the greater net precipitation in summer in $MH_{GS}$ is

largely compensated by reduced snowmelt.

### 4.3 The UBB

The Brahmaputra River originates from a glacier in the Himalayas at an elevation of 5.8 km and passes through the southern TP before it breaks through the eastern Himalayas and turns southward. Elevation of the UBB, above the Yangcun hydrometric station ranges between 5.8 km and 3.8 km and drops off from the west to the east. Note that the UBB lies in a

wetter region than the UYEB and UYAB, and the average precipitation rate is much higher (Fig. 3a), so that the basin-integrated water flux is also larger.

The seasonal cycle of hydroclimatic variables from the GCM and the WRF ensemble for the UBB is shown in Fig. 11. Based on the observation, the basin-averaged annual precipitation is 741 mm and the annual mean temperature is −1 °C. The global model has a year-round cold bias, which is particularly severe in winter. The general representation of the temperature

seasonal cycle is significantly improved in the WRF ensemble where the DFJ cold bias is reduced by half. However, the



WRF ensemble has a larger cold bias than the GCM in spring, which is likely associated with the poor representation of snowmelt processes by the much simpler snow model employed in WRF. The winter precipitation is well capture in the WRF ensemble, while summer precipitation is somewhat underestimated. The global model is, however, characterized by a wet bias in all seasons. Note here that the station density in parts of the UBB is very low, so that the observational error can

also be large (Harris et al., 2020). All of the precipitation from November to April falls in solid form, which leads to a snowmelt peak in late spring and early summer. Unlike UYEB and UYAB, snowfall in UBB is more evenly distributed throughout the year. In observations as well as in our simulations, total precipitation starts to increase in June, but net precipitation is small in early summer due to evapotranspiration (Fig. 11c). Net precipitation from the first WRF ensemble member with the KF cumulus scheme is in fact negative in May and June. However, runoff over the UBB starts to increase

in May (Fig. 11d), which possibly comes from the snowmelt contribution that is the largest in May and June (Fig. 11c). In UBB, the JJAS flow accounts for approximately 70% of the annual total according to observations obtained at the Yangcun station. The peak flow in the UBB occurs in August, consistent with the highest net precipitation in July and August, which indicates that UBB rainfall plays a dominant role in annual total surface runoff. The underestimation in the simulated peak streamflow is most likely due to the underestimation of precipitation over the basin. The first WRF ensemble member, which

produces the greatest runoff underestimation, also has the largest negative bias in JJAS precipitation.

During the MH, annual mean temperature is simulated to drop by approximately 0.7 °C in the WRF ensemble in response to orbital forcing alone (Fig. 12a) and rise by 0.7 °C in experiments that account for a vegetated Sahara (Fig. 12b). UofT-CCSM4 also shows a similar shift from negative to positive temperature anomalies when the greening of the Sahara is taken into consideration. In both WRF MH experiments, the warming appears to be more pronounced at the end of the monsoon

season (September and October). The average JJAS warming is 0.5 °C in $MH_{REF}$ and 1.5 °C in $MH_{GS}$. On average, annual precipitation in the WRF ensemble increases by approximately 0.15 mm $d^{-1}$ for $MH_{REF}$ (i.e. 11% increase with individual ensemble members ranging from 3%–17%) and 0.35 mm $d^{-1}$ for $MH_{GS}$ (i.e. 26% increase with an ensemble range of 19%–37%). Both MH experiments show a significant increase in precipitation in JJAS after a small decrease in late winter–early spring and little change in precipitation is seen in November and December. Compared to the increase in total precipitation,

the change in the total amount of snow is very small for both MH experiments (< 0.04 mm $d^{-1}$), which also leads to very small changes in the annual snowmelt. A major change in the basin hydrology is the increase of net precipitation in JJAS, which also results in an overall increase in runoff. In $MH_{REF}$, streamflow first decreases slightly in May and June and then increases in the second half of the year with a peak in July. In WRF simulations that consider the influence of the Sahara greening, runoff still exhibits increasing trends between July and September, which is consistent with changes in

precipitation, but little change is detected for runoff in the first half of the year. The overall river discharge increasing trend over the UBB is robust across the WRF ensemble (~27% in ensemble average of $MH_{REF}$ and ~55% in $MH_{GS}$), but the magnitude of the increase is subject to higher model uncertainties than the other two basins, which is, however, not unexpected: the UBB is mostly strongly influenced by the Asian monsoon and features the strongest JJAS rainfall and the largest annual discharge. Thus, different cumulus schemes have a large impact on the changes in annual total surface runoff.





Similar river runoff increase has also been found in future projections in a warmer climate using a hydrological model and multi-source spatial data (Cai et al., 2017) and LIG hydrological simulations based on PMIP4 data (Scussolini et al., 2020).

## 5 Summary and conclusions

We have presented dynamically downscaled high-resolution climate simulations for the TP, with an analysis of the hydroclimate in three major river basins, the UYEB, UYAB and UBB. The fully coupled global climate model UofT-
CCSM4 was used to generate climate simulations for the modern and MH time periods, which were dynamically downscaled to a resolution of 10 km, using coupled WRF-WRF-Hydro with four different cumulus parameterization schemes. Compared to the GCM, WRF is far superior in representing orographic precipitation and the variation across the seasonal cycle. It captures the sharp contrast between the very high precipitation rates south of the Himalayan Mountain ranges and the dry climate in their rain shadows over the interior plateau. Both UofT-CCSM4 and WRF configurations suffer from a cold bias
in winter due to excessive snowfall, but the regional model has again significantly reduced this cold bias.

The coupled UofT-CCSM4-WRF-WRF-Hydro model has also been used to investigate the changes in temperature and precipitation over the TP during the MH. Both the GCM and the WRF ensemble average show a warming over the TP during the boreal summer and a cooling in winter, which results from insolation change. As far as the JJAS temperature is concerned, the WRF simulations show a larger warming than that in the driving global model, especially in the northern part
of TP, and is thereby closer to estimates from pollen records (Shi et al., 1993). The enhanced land-sea contrast during summer favours the strengthening of the monsoon circulation, featuring stronger southerly winds from the sea, which in turn enhances the northward water vapour transport. The general spatial patterns for the changes of TP precipitation are similar in sign between the GCM and the RCM, with significant increase in the southern parts of the TP. The seasonal cycle of rainfall over the TP is modified by the change of insolation, such that precipitation weakens slightly in spring and strengthens during
the summertime. The annual wet anomalies are stronger and more extensive in the downscaled simulations and thus fit better with the proxy data. Since the TP plays a profound role in the development of the Asian monsoon and heavy precipitation forms when the wind from the sea rises along the south and east edges of the plateau, the model's ability to realistically represent the complex steep topography may have a great impact on the coupling between the changes over the TP and the general monsoon circulations. Including a green Sahara further strengthens the TP precipitation and halves the bias against
reconstructions over the TP, which highlights the significance of the teleconnections between the Saharan vegetation and the TP in paleoclimate simulations. Precipitation changes in MH experiments which assume preindustrial vegetation cover are to a good approximation about half the magnitude of changes in MH$_{GS}$ in all WRF ensemble members, although there are notable differences in the simulated MH rainfall anomalies using different physics schemes.

UYEB hydrological regimes exhibit changes in the MH as manifested by decreases in annual streamflow. The largest
reduction occurs in September in MH$_{GS}$, one month later than in MH$_{REF}$. These changes in the simulated streamflow were determined to be the combined effects of changes in rainfall and evapotranspiration. The JJAS net precipitation is simulated





to shrink by about 30% in MH$_{REF}$ and 10% in MH$_{GS}$. Snowmelt also plays a modest role in UYEB hydrology. The GS forcing caused a rise in temperature during the MH, as well as larger rainfall but smaller snowmelt compared to the MH$_{REF}$.

The most important change in the MH in the UYAB is a significant shift from solid to liquid precipitation. Both MH

experiments show a decrease in annual snowmelt (17% in MH$_{REF}$ and 19% in MH$_{GS}$, with ±10% variability between ensemble members); a warming of 1.3 ∘C is simulated in MH$_{GS}$ while a small annual cooling is shown in the WRF experiments without the GS boundary forcing. The standard MH experiments produce a 9% annual precipitation decrease, while MH$_{GS}$ shows a small increase of approximately 4%. However, the net effect of all these changes in both MH experiments is a large reduction in streamflow, especially in summer.

In the UBB the simulated annual total precipitation increase in the MH is the largest among three basins: about 11% in the ensemble mean in MH$_{REF}$ and 26% in MH$_{GS}$. The simulated JJAS warming is 0.5 ∘C in MH$_{REF}$ and 1.5 ∘C in MH$_{GS}$, but, unlike the UYAB, the amount of solid precipitation changes little because the amount of snowfall in PI is much smaller than that in UYAB. Based on the simulations presented here, the most significant MH hydroclimatic anomaly in the UBB may be an increased runoff (~27% in MH$_{REF}$ and ~55% in MH$_{GS}$), especially in mid-summer, because of greater net precipitation

and runoff generation.

A fundamental uncertainty limiting the validity of the MH simulations presented here is natural variability, including the El Niño–Southern Oscillation. This clearly underscores the need for an initial condition ensemble where the RCM is forced with different periods of the global simulations. Besides, this study has used only one coupled regional modeling system to investigate the influence on the TP climate only due to the MH land cover changes over northern Africa. However, some

recent studies also argue for an impact on the MH Asian monsoon resulting from reduced dust over northern Africa (Sun et al., 2019; Pausata et al., 2020). Additionally, reconstructions indicate widespread vegetation changes during the MH, particularly greater vegetation coverage over Eurasia (Tarasov et al., 1998; Zhang et al., 2014), which has been shown to be able to shift the intertropical convergence zone northward and have a global impact (Swann et al., 2014). These results call for further studies using both different numerical models and more realistic dust concentrations and vegetation distributions

in the paleoclimate simulations over the TP.

**Code/Data availability**

Code and data for reproducing each of the figures in the paper can be obtained from Yiling Huo.

**Author contribution**

WRP and YH designed the experiments and YH carried them out. DC designed the Green Sahara simulation and performed

the UofT-CCSM4 global experiments. YH adapted the WRF radiation module for the MH experiments with the help of DC.



YH modified the dynamical downscaling pipeline to include WRF-Hydro. YH prepared the manuscript with contributions from both co-authors.

**Competing interests**

The authors declare that they have no conflict of interest.

**Acknowledgments**

The simulations presented in this paper were performed at the SciNet High Performance Computing facility at the University of Toronto, which is a component of the Compute Canada HPC platform.

**Financial support**

This research has been supported by the Natural Sciences and Engineering Research Council of Canada (grant no. A9627).

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




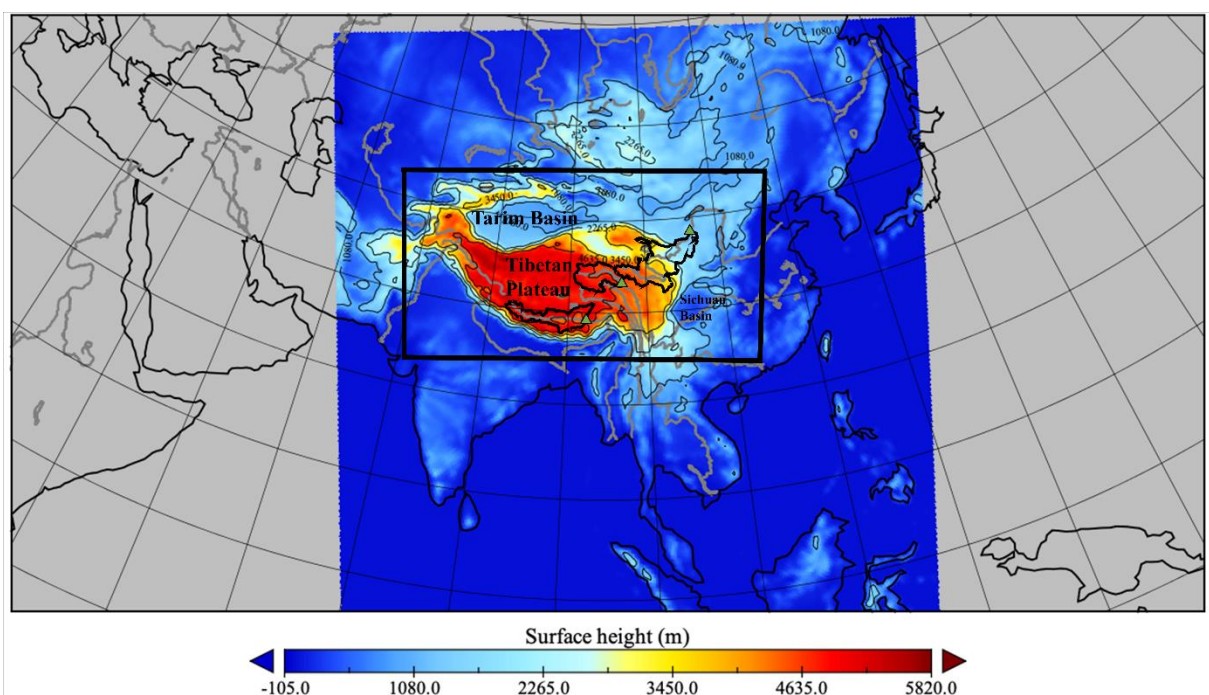

**Figure 1: Topography (m) and outlines of the outer and inner WRF domains, as well as the major rivers and lakes (grey). Black curves denote outlines of the UBB, UYAB and UYEB at the left, middle and right. Green triangles inside each basin denote the Yangcun (left), Zhimenda (middle) and Tanglai Qu (right) stations.**

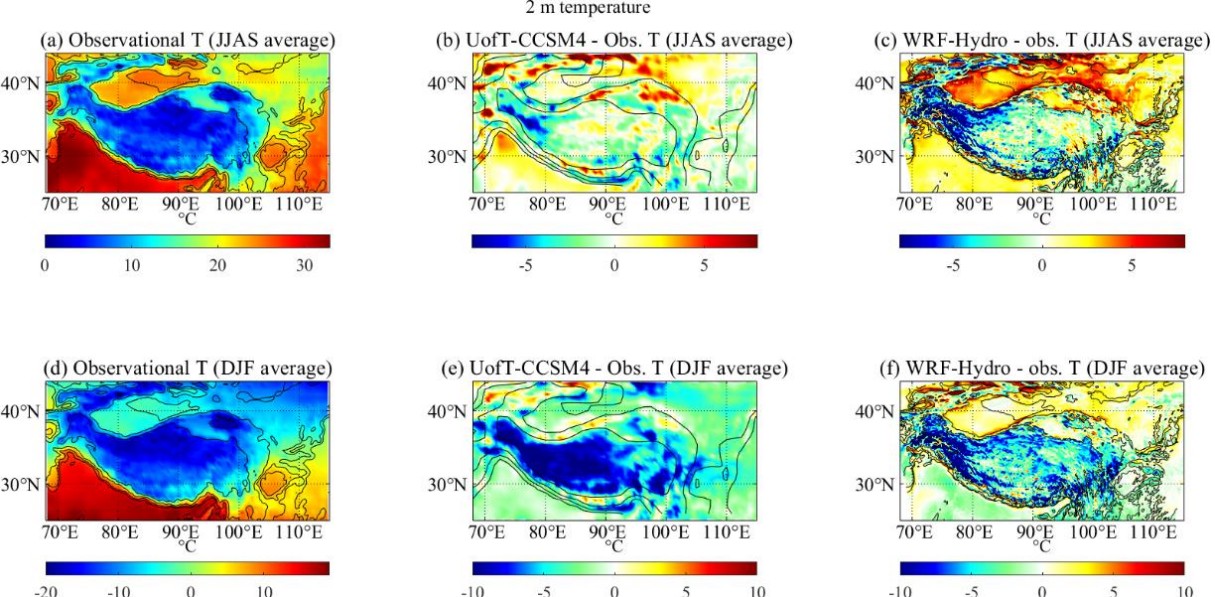

**Figure 2: Absolute 2-m air temperature bias with respect to (a, d) the CRU observational dataset for (b, e) the UofT-CCSM4 and (c, f) the WRF ensemble in (a, b, c) JJAS and (d, e, f) DJF. The topography contours of 500, 1000, 2000, and 4000 m are also shown in black.**



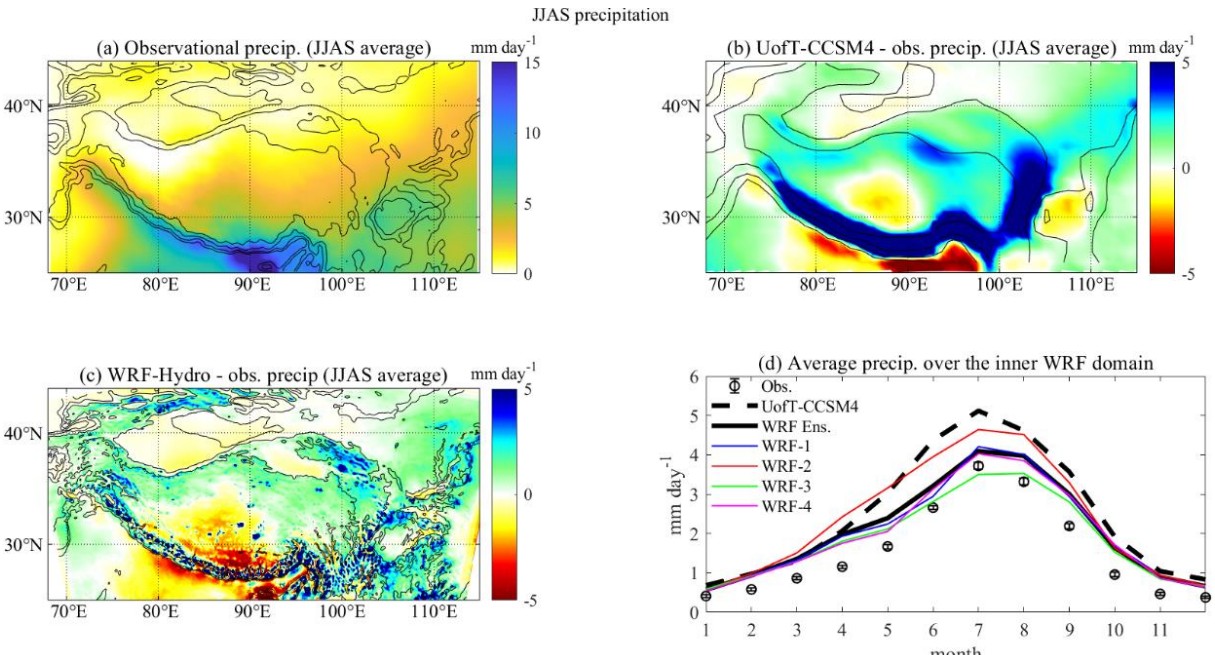


**Figure 3: JJAS precipitation bias with respect to (a) the CRU observational dataset for (b) the UofT-CCSM4 and (c) the WRF ensemble. (c) Monthly precipitation over the inner WRF domain in the observation, UofT-CCSM4, WRF ensemble average and four individual physics ensemble members. The topography contours of 500, 1000, 2000, and 4000 m are also shown in black in (a, b, c).**

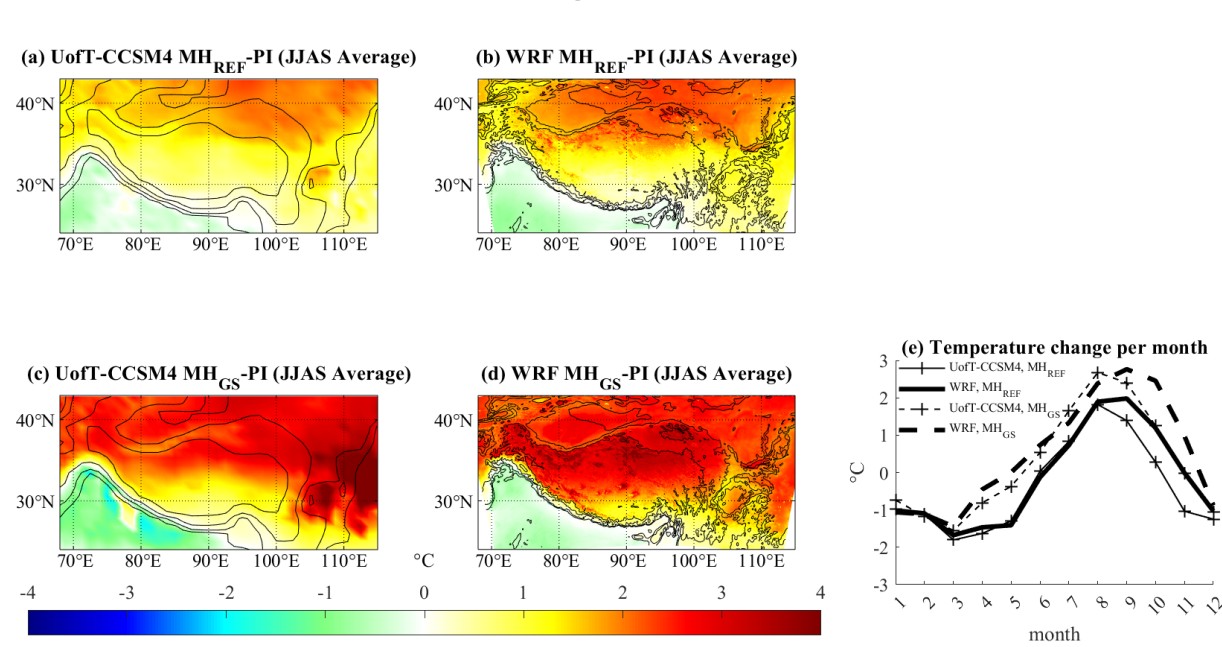


**Figure 4: JJAS surface air temperature anomalies (◦C) for (a, b) MH_REF and (c, d) MH_GS in (a, c) UofT-CCSM4 and the (b, d) WRF ensemble mean. (e) Monthly continental air temperature anomalies for MH_REF (solid) and MH_GS (dashed). Shifts in calendar**





are not accounted for; i.e., the model calendar is used for the calculation of all anomalies. The topography contours (black) of 500, 1000, 2000, and 4000 m are also shown in (a-d).

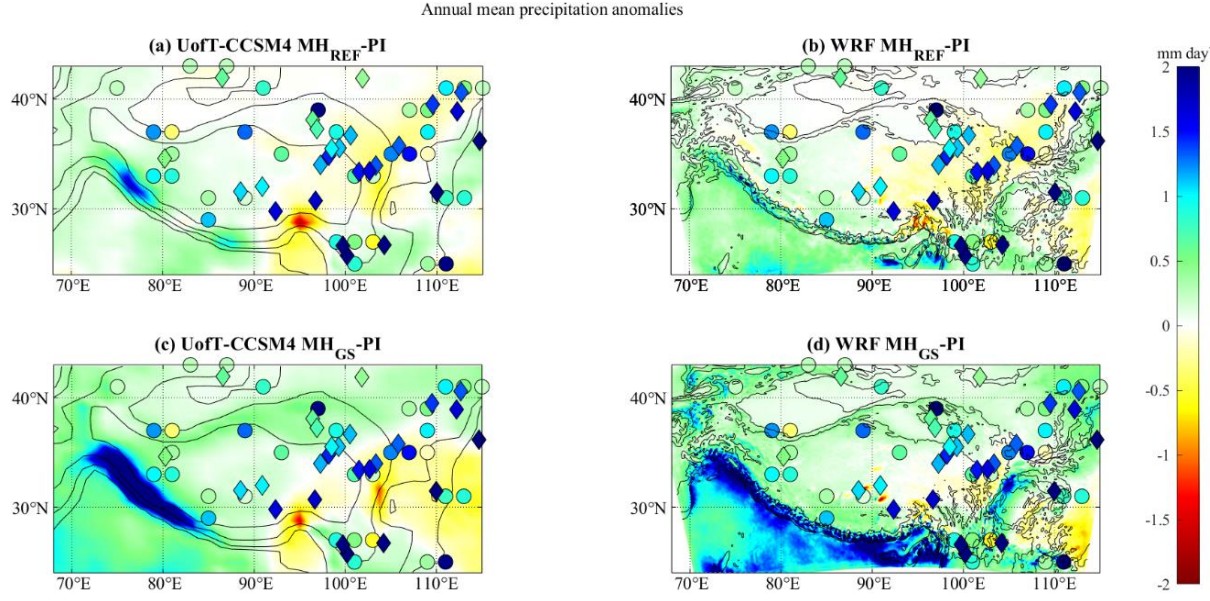

**Figure 5:** Annual mean precipitation anomalies (mm d$^{-1}$) for (a, b) MH$_{REF}$ and (c, d) MH$_{GS}$ in (a, c) UofT-CCSM4 and the (b, d) WRF ensemble mean. Reconstructed precipitation differences between the MH and present day from Bartlein et al. (2011) and Herzschuh et al. (2019) are plotted as circles and diamonds, respectively. The topography contours of 500, 1000, 2000, and 4000 m are also shown.

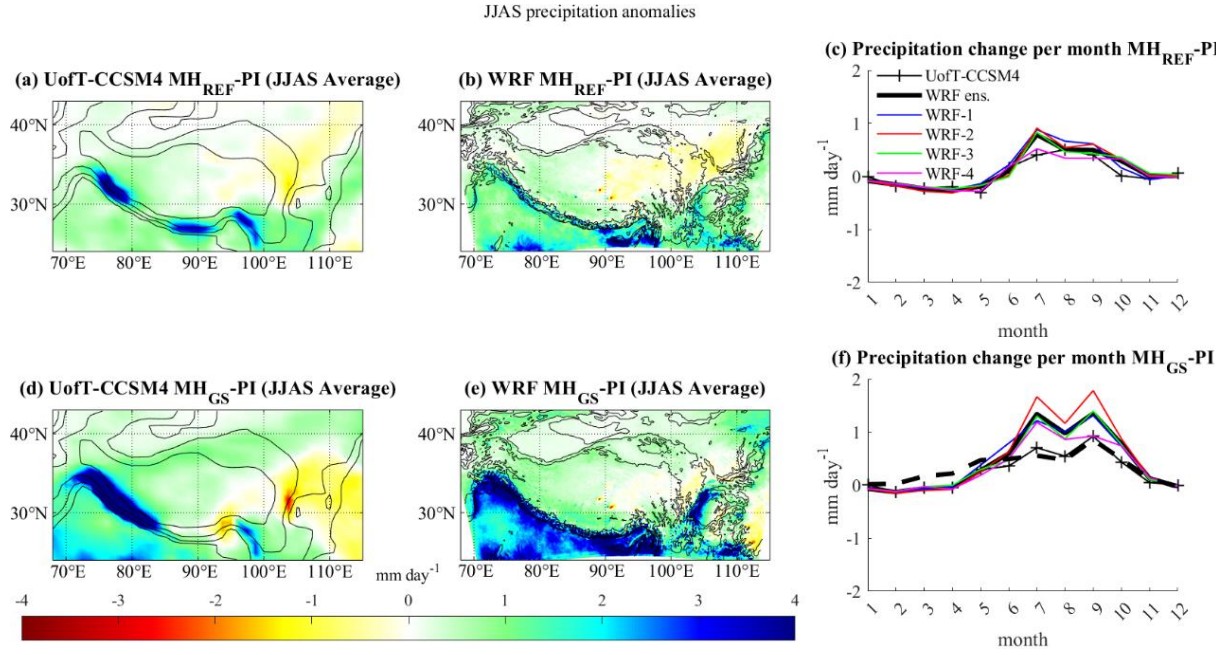

**Figure 6:** JJAS precipitation anomalies (mm d$^{-1}$) for (a, b) MH$_{REF}$ and (d, e) MH$_{GS}$ in (a, d) UofT-CCSM4 and the (b, e) WRF ensemble mean. Monthly precipitation anomalies (mm d$^{-1}$) in UofT-CCSM4 and four physics ensemble members over for (c)






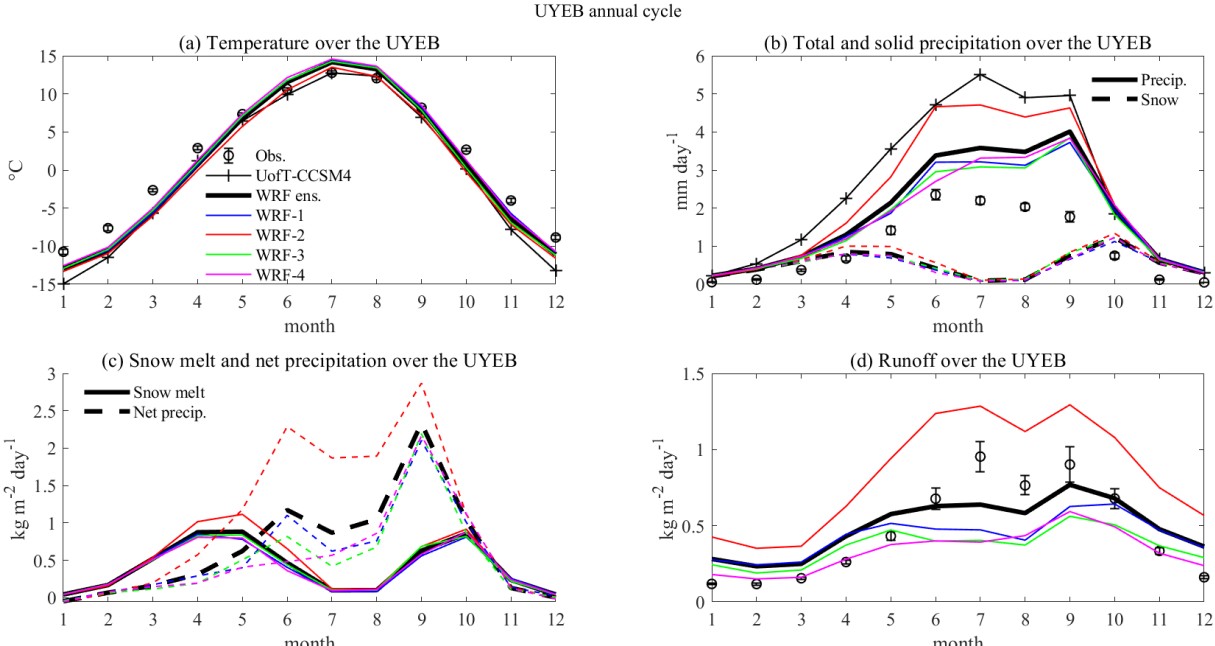

MH$_{REF}$ and (d) MH$_{GS}$. Shifts in calendar are not accounted for; i.e., the model calendar is used for the calculation of all anomalies. The dashed line in (f) represent the MH$_{GS}$–MH$_{REF}$ anomalies in the WRF ensemble mean. The topography contours of 500, 1000, 2000, and 4000 m are also shown in (a, b, d, e).

**Figure 7: Average seasonal cycle over the UYEB: (a) 2-m air temperature; (b) total and solid precipitation; (c) net precipitation and snowmelt and (d) discharge at the hydrometric station and total runoff from WRF. All quantities are averaged over the entire basin; river discharge was normalized by the basin area. The WRF ensemble average is shown in thick black lines, four individual ensemble members are shown by blue, red, green and magenta lines, respectively; observed values are indicated with circles. Error bars/bands show the standard error of the mean at a 95% confidence level.**






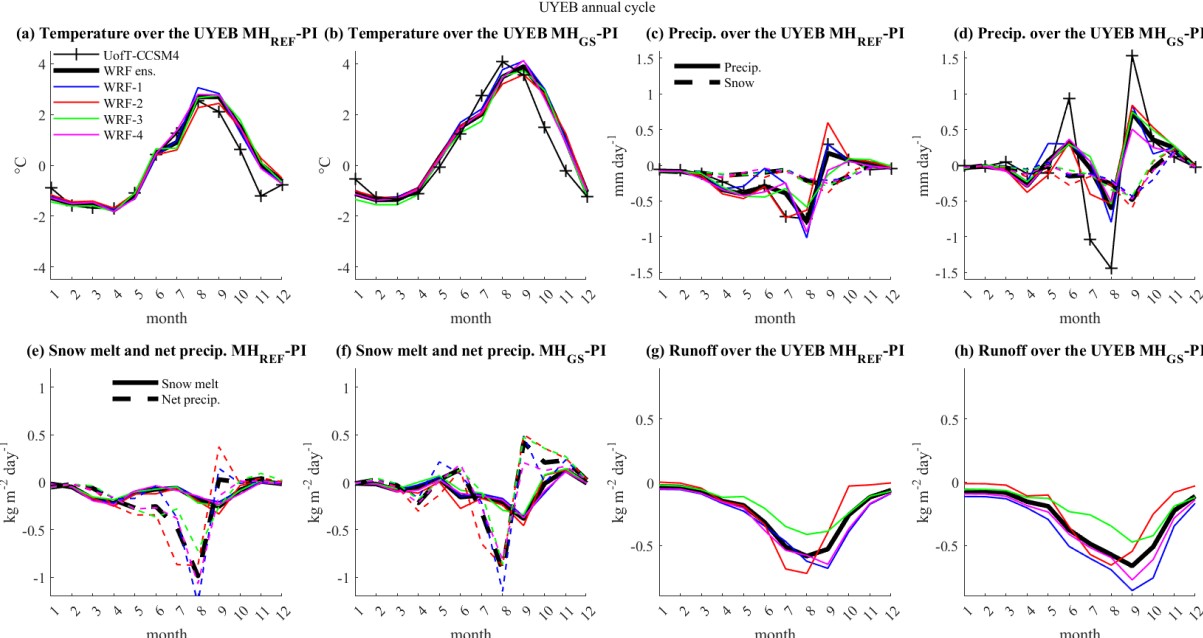

**Figure 8: Monthly anomalies over the UYEB for (a, c, e, g) MH$_{REF}$ and (b, d, f, h) MH$_{GS}$ of: (a, b) 2-m air temperature; (c, d) total and solid precipitation; (e, f) net precipitation and snowmelt and (g, h) river runoff. All quantities are averaged over the entire basin.**

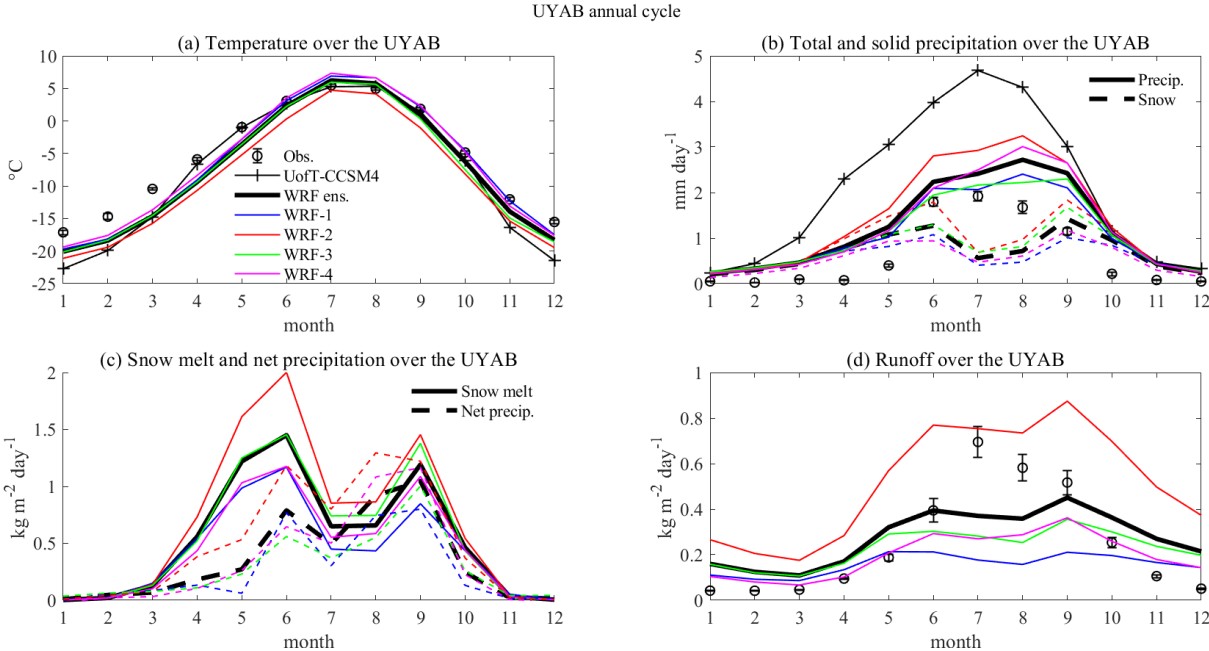

**Figure 9: As in Fig. 7, but for the UYAB.**





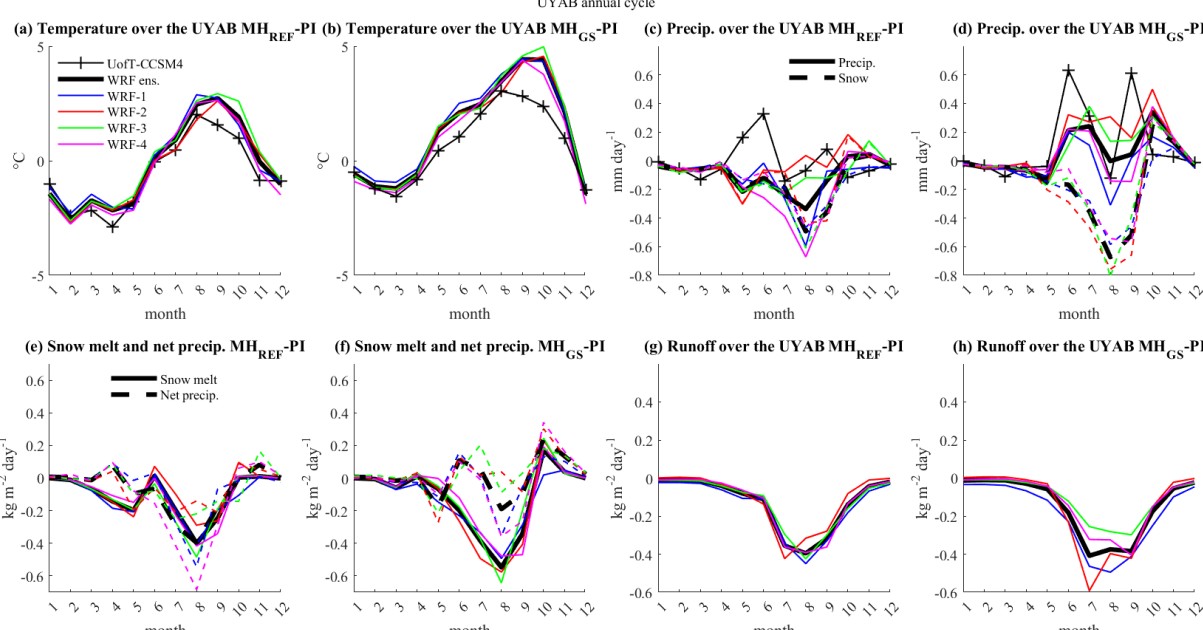

**Figure 10: As in Fig. 8, but for the UYAB.**

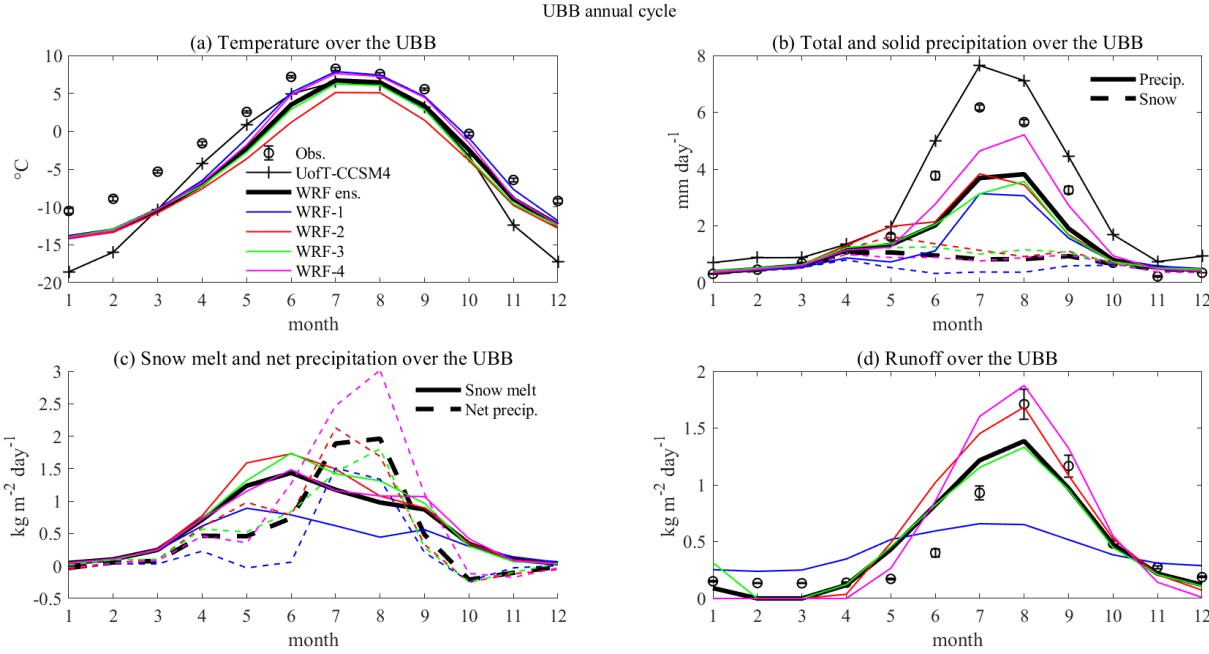


**Figure 11: As in Fig. 7, but for the UBB.**





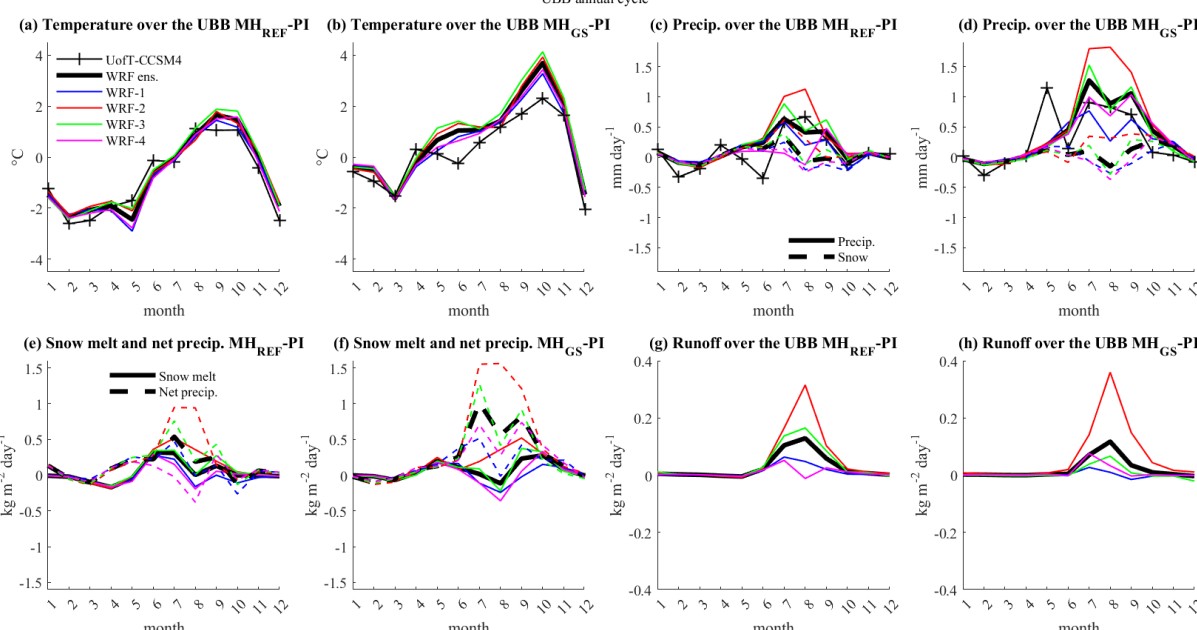

**Figure 12: As in Fig. 8, but for the UBB.**