# Peer review of "Mid-Holocene climate of the Tibetan Plateau and hydroclimate in three major river basins based on high-resolution regional climate simulations"

_EGUsphere, 2022_

## Author Response (AR1)

**Response to the Comments of Editor on the paper "Mid-Holocene climate of the Tibetan Plateau and hydroclimate in three major river basins based on high-resolution regional climate simulations" by Yiling Huo, W. R. Peltier and Deepak Chandan**

We thank the editor for his valuable comments on the content of our manuscript and his suggestions for improving the document. Following the editor's suggestions and comments, we have carefully revised our manuscript. We believe that the revised version satisfactorily addresses the editor's questions and concerns. In this reply, we respond to the issues, raised by the editor point by point. Our responses to the individual comments are shown in red text following the comments in black. For convenience, the modifications made to the text will also be shown in red.

Dear authors,

Based on the positive reviews and your thorough reply to the reviewers, I invite you to send a revised version of their manuscript. The manuscript will be reviewed again, as requested by the reviewers.

I would like to take the opportunity to add a few minor comments which you might consider.

a) What is a "fully" coupled model? I suggest to replace this meaningless term, which one can often find in the literature, by a more specific term like synchronously coupled, for example. Or simple skip "fully".

Thank you for this suggestion. We have now changed "fully" to 'synchronously' as suggested, which you will see in line 115.

b) I am curious whether the Toronto-CCSM shows the same cumbersome precipitation bias over Sahara/Sahel as the CCSM 4 or more precisely, the CAM 4 does (see https://cp.copernicus.org/articles/18/313/2022/ ). In the later model version, the isohyets over the Sahel reveal an unrealistic west-east gradient with a moister eastern Sahara and Arabia peninsula. This could have an effect on the precipitation downstream of Arabia.

We are not sure what the editor meant by the "cumbersome precipitation bias over Sahara/Sahel". Did the editor mean there is a bias in the PI simulation or the MH-PI wet anomaly is overestimated over the eastern Sahara and Arabia peninsula? If the latter, the precipitation increase in Western Sahara in our global model is actually greater than Eastern Sahara although there is also a wet anomaly core over the western Arabian Peninsula where the Asir Mountains lead to significant orographically forced precipitation (Fig. 7). Also, the paper the editor pointed to (https://cp.copernicus.org/articles/18/313/2022/) mainly addressed the impact of soil biophysics and soil nitrogen on the simulation of the GS, and didn't really mention an unrealistically moist eastern Sahara and Arabia peninsula.

c) Pausata et al. (2017) did not simulate the feed-back between a green Sahara and the Asian monsoon, but the impact of a prescribed pretty green Sahara and a strong dust effect on the Asian monsoon.

We have adjust the text in lines 89-91 to "Studies have shown that a vegetated Sahara and the strengthened African monsoon can enhanced the Asian monsoon by altering the Walker circulation through changes in tropical Atlantic SSTs (Pausata et al., 2017)." to make it more accurate.

d) A word on the prescribed vegetation patterns used in the global simulation would be useful. It seems that complete coverage and likely unrealistic - but who knows – zonal vegetation patterns are considered based on the early reconstruction by Hoelzmann et al. (1998). This choice also might affect the results of the study.

We have now added in lines 130-134:

"The prescribed vegetation patterns used in the $MH_{GS}$ experiment are similar to those used by Pausata et al. (2016) and are consistent with the proposed MH vegetation sensitivity experiments in Otto-Bliesner et al. (2017), which involves a substitution of the Sahara desert with evergreen shrubs in the south and steppe and savanna in the north. Moreover, based on the results of Hély et al. (2014), the African Guineo-Congolian rain forest has also been extended slightly northward (Chandan and Peltier, 2020)."

Hély, C., Lézine, A. M., and Contributors, A. P. D.: Holocene changes in African vegetation: Tradeoff between climate and water availability, Climate of the Past, 10(2), 681–686, https://doi.org/10.5194/cp-10-681-2014, 2014.

Pausata, F. S. R., Messori, G., and Zhang, Q.: Impacts of dust reduction on the Northward expansion of the African monsoon during the Green Sahara period, Earth and Planetary Science Letters, 434, 298–307. https://doi.org/10.1016/j.epsl.2015.11.049, 2016.

Best regards,

Martin Claussen

**Response to the Comments of Referee 1 on the paper "Mid-Holocene climate of the Tibetan Plateau and hydroclimate in three major river basins based on high-resolution regional climate simulations" by Yiling Huo, W. R. Peltier and Deepak Chandan**

We thank the referee for his/her valuable comments on the content of our manuscript and his/her suggestions for improving the document. Following the reviewer's suggestions and comments, we have carefully revised our manuscript. We believe that the revised version satisfactorily addresses the referee's questions and concerns. In this reply, we respond to the issues, raised by the referee point by point. Our responses to the individual comments are shown in red text following the comments in black. For convenience, the modifications made to the text will also be shown in red.

In this manuscript, Huo and coauthors conducted a series of dynamically downscaled high-resolution simulations to analyze hydroclimate responses over Tibetan Plateau (TP) under the Pre-industrial (PI) and mid-Holocene (MH) conditions with and without a green Sahara condition. In particular, results from a fully coupled global-scale climate model (the University of Toronto version of CCSM4) are downscaled to 10 km resolution using four different cumulus parameterization schemes in the Weather Research and Forecasting Model coupled with the hydrological model WRF-Hydro. The authors made great efforts to reproduce characteristics of the TP's hydroclimate in the WRF-Hydro of which spatial resolution is competent in representing orographic impacts on precipitation and its seasonal variability. However, the validation against historical observation and the demonstration of MH climate is insufficient. Nevertheless, the study could be the first step to simulate MH-TP hydroclimate change in a high-resolution regional climate model, and hence I recommend acceptance for publication after considering the following comments.

Many thanks to the reviewer for this overall positive feedback, which we appreciate. We agree that the readers will likely find our paper new and interesting. We have also taken care to strengthen the validation against historical observation and the demonstration of MH climate.

Major comment:

The land surface has significant impact on climate and hydrology. For example, Yue et al. (2021) found that different types of underlying surfaces affect the partitioning of sensible and latent heat fluxes, causing different local circulations and further impacting precipitation and temperature over the southern TP. Implementation of more accurate soil texture can lead to reduced biases in simulated soil moisture and impact simulated runoff and evaporation (De Lannoy et al., 2014). In the manuscript, the same land surface on the TP was used in both PI and MH simulations. During the Holocene, Chen et al. (2020) revealed that the maximum forest extent was reached in the MH. That may have some impact on climate and hydrology. To some extent, the evolution process of vegetation on TP should be considered. Li et al. (2019) has already reconstructed pattern of vegetation evolution for China since the Last Glacial Maximum by pollen dataset. Therefore, given the main goal of this study, it is necessary to consider changes in the land surface of TP during the MH too.

Yue S, Yang K, Lu H, et al. Representation of Stony Surface–Atmosphere Interactions in WRF Reduces Cold and Wet Biases for the Southern Tibetan Plateau. Journal of Geophysical Research: Atmospheres, 2021, 126(21): e2021JD035291.

De Lannoy G J M, Koster R D, Reichle R H, et al. An updated treatment of soil texture and associated hydraulic properties in a global land modeling system. Journal of Advances in Modeling Earth Systems, 2014, 6(4): 957-
979.

Chen F, Zhang J, Liu J, et al. Climate change, vegetation history, and landscape responses on the Tibetan Plateau during the Holocene: a comprehensive review. Quaternary Science Reviews, 2020, 243: 106444.

Li Q, Wu H, Yu Y, et al. Large-scale vegetation history in China and its response to climate change since the Last Glacial Maximum. Quaternary International, 2019, 500: 108-119.

We agree with the referee that the land surface has significant impact on climate and hydrology, but the purpose of our paper is to isolate the impact of the Green Sahara on the hydroclimate impacts on the TP. Thus, adding the regional impacts of surface vegetation changes in China on MH hydroclimate changes on the plateau would detract from the main focus of this study. Furthermore, the suggested data set for the mid-Holocene vegetation of China (Li et al., 2019) seems not to be publically available online yet. However, we still added references to Chen et al.
(2020) and Li et al. (2019) in the last paragraphy, where our original manuscript already addressed the influences of Eurasian forests during the MH and proposed possible future work.

Minor comments:

1. The definition of TP in the Introduction is inconsistent with the WRF inner domain in the main text, which is misleading. Please clarify this as well as the relationship between the TP and the WRF inner region.

It is true that definition of the TP and our WRF inner domain are not the same, and we have already stated in lines 135 that "while the inner domain encompasses the TP, as well as parts of the surrounding territory". We have now added ", which covers the TP, as well as some surrounding regions" at the beginning of our results section in line 183 to make it clearer.

2. The authors provided a proper data-model comparison regarding precipitation to assess the performance of
the experiments. Since there is also abundance of temperature records in the studied area and temperature is an important atmospheric parameter for hydroclimate (Zhang et al., 2022; Kaufman et al., 2020), it is necessary to include it in the comparison.

Zhang C, Zhao C, Yu S Y, et al. Seasonal imprint of Holocene temperature reconstruction on the Tibetan Plateau. Earth-Science Reviews, 2022: 103927.

Kaufman D, McKay N, Routson C, et al. A global database of Holocene paleotemperature records. Scientific data, 2020, 7(1): 1-34.

Thank you for suggesting we compare our simulated temperature with proxy data. We've added comparison to temperature records from Kaufman et al. (2020) and Zhang et al. (2022) in Fig. 4 and lines 227-239:

"Moreover, compared to paleoclimatic reconstructions based on lake sediment in Fig. 4 (Zhang et al., 2022;

Kaufman et al., 2020), both MH experiments generally capture the summer warming trend over the TP. In south-eastern TP, there is a good agreement between the model simulations and the reconstructions. The simulated temperature anomalies in $MH_{GS}$ fit the reconstructed paleoclimatic records in the north-eastern and south-central TP better but overestimate the warming signal over the central-eastern part of the TP (Fig. 4d). Meanwhile, the two proxy data points located in the central-western TP west of the 85° W indicate strong cooling ($< -4$ °C) during the MH, which disagrees with all model simulations. Note here these two records are from frozen lakes, where the reconstructed temperature records reveal air temperature changes during the ice-free season (May - September), not just JJAS. Also note here all proxy records are subject to uncertainties that arise from the dating processes (Wang et al., 2021) and site-specific factors like the change in vegetation cover and the retreat of glaciers (Chen et al., 2020a). Averaged over all the points except these two, both UofT-CCSM4 and the WRF ensemble average have a bias of $-1.2$ °C in $MH_{REF}$. Inclusion of a GS greatly reduces this cold bias to $-0.1$ °C in WRF, which is also smaller than that of the global model in $MH_{GS}$ ($-0.4$ °C)."

Chen, X., Wu, D., Huang, X., Lv, F., Brenner, M., Jin, H., Chen, F.:Vegetation response in subtropical southwest China to rapid climate change during the Younger Dryas, Earth Sci. Rev., 201, p. 103080, https://doi.org/10.1016/j.earscirev.2020.103080, 2020a.

Wang, M., Hou, J., Duan, Y., Chen, J., Li, X., He, Y., Lee, S. Chen, F.: Internal feedbacks forced Middle Holocene cooling on the Qinghai-Tibetan Plateau, Boreas, 50, 1116-1130, https://doi.org/10.1111/bor.12531, 2021.

[Figure]

[Figure]

**Figure 4: JJAS surface air temperature anomalies (∘C) for (a, b) MHREF and (c, d) MHGS in (a, c) UofT-CCSM4 and the (b, d) WRF ensemble mean. (e) Monthly continental air temperature anomalies for MHREF (solid) and MHGS (dashed). Reconstructed temperature differences between the MH and present day from Zhang et al. (2022) and Kaufman et al. (2020) are plotted as circles and diamonds, respectively. The topography contours (black) of 500, 1000, 2000, and 4000 m are shown in (a-d).**

3.  There is no doubt that WRF is competent in simulating regions with complex terrain than GCM. However, the biases between WRF and observation are obvious in the simulation of temperature and precipitation. Authors shall adequately discuss this weakness and its potential role in their results.

We have now added some discussion regarding the influence of temperature and precipitation biases in winter. In lines 189-190, 194-195 and 206-209, we now state:

"Most of the UYEB and UYAB are affected by this warm bias, which will likely reduce summer streamflow due to increased evapotranspiration."

"Among the three river basins, the UBB is most strongly affected by the cold bias in winter, which may decrease streamflow in the cold season due to late snowmelt."

"The overestimation in precipitation over the western and south-eastern TP in WRF is also accompanied by lower surface temperature, which is likely related to the greater cloud cover reflecting more shortwave flux at high levels. Both models show a similar wet bias in winter (Fig. 3d) and such excessive snow in winter possibly contributes to the lower DJF temperature (Fig. 2) through snow–albedo feedback."

4. The results in Section 4 are too detailed and unfocused, it is hardly to catch the points. Can you shorten the results to be more readable? Section 5 also exists the same problem. Please briefly summarize the conclusions.

We have reduced section 5 by around 15%. We have also restructed section 4 and shortened it by around 1/3.

160 5. The CRU dataset is selected as observation dataset to verify the results in historical period (1980-1994). However, it is hard to say that CRU has a well performance in describing the precipitation on the TP. A more convincing dataset or evidence showing the validation of the CRU should be mentioned in the manuscript.

Thank you for this suggestion. We now instead use APHRODITE's (Asian Precipitation - Highly-Resolved Observational Data Integration Towards Evaluation) gridded precipitation dataset version 1101 (Yatagai et al,

165 2012), which covers Monsoon Asia (APHRO_MA_V1101) at 0.5∘×0.5∘ horizontal resolution. It is a set of long-term (1951 onward) continental-scale daily products based on a dense network of rain-gauge data for Asia including the Himalayas, South and Southeast Asia. However, using this new dataset didn't significant change the results in our original manuscript (Fig. 3). Comparison of the model results and CRU precipitation in JJAS during the historical period has now been moved to Fig. A1.

170 Yatagai, A., Kamiguchi, K., Arakawa, O., Hamada, A., Yasutomi, N., and Kitoh A. APHRODITE: Constructing a Long-term Daily Gridded Precipitation Dataset for Asia based on a Dense Network of Rain Gauges, Bulletin of American Meteorological Society, https://doi.org/10.1175/BAMS-D-11-00122.1, 2012.

[Figure]

**Figure 3: JJAS precipitation bias with respect to (a) the APHRODITE observational dataset for (b) the UofT-CCSM4 and (c) the WRF ensemble. (c) Monthly precipitation over the inner WRF domain in the observation, UofT-CCSM4, WRF ensemble average and four individual physics ensemble members. The topography contours of 500, 1000, 2000, and 4000 m are also shown in black in (a, b, c).**

6.  Lines 99 and 221: Please confirm the expression "and. Since" and "half. Reconstruction".

We apologize for the error in line 99, and we have corrected the text by removing "and". There was no error in line 221, but we have still adjusted the text from "further reducing the bias with respect to the reconstruction by Bartlein et al. (2011) by half." to "further reducing the bias with respect to the pollen-based reconstructions (Bartlein et al., 2011) by half" to be clearer.

7.  Rewrite the last sentence of Abstract.

We have now rewritten the last sentence in the abstract:

"The simulation results were first validated over the upper basins of the three rivers before the hydrological responses to the MH forcing for the three basins were quantified. Both the upper Yellow and Yangtze rivers exhibit a decline in streamflow during the MH, especially in summer, which is a combined effect of less snowmelt and stronger evapotranspiration. The GS forcing caused a rise in temperature during the MH, as well as larger rainfall but less snowfall and greater evaporative water losses. The Brahmaputra River runoff is simulated to increase in the MH, due to greater net precipitation."

8.  Lines 106 and 212: Please show the full name before using the abbreviation.

We apologize for these errors, and we have corrected the text by stating "upper Yellow, Yangtze and Brahmaputra River basins (UYEB, UYAB and UBB, respectively)" in line 106 and using the full name "greenhouse gas" instead of "GHG" in line 212.

9.    Lines 145-147: The description way is weird here. Can you give a better way? For example, the dynamical downscaling methodology employed here is a somewhat further developed version of the dynamical downscaling "pipeline" originally introduced in Gula and Peltier (2012) and then widely applied in recent studies.

Thank you for this suggestion. We have now changed the text as suggested.

10.   Gula J, Peltier W R. Dynamical downscaling over the Great Lakes basin of North America using the WRF regional climate model: The impact of the Great Lakes system on regional greenhouse warming. Journal of Climate, 2012, 25(21): 7723-7742.

We are not quite sure what the reviewer meant by this comment. This paper is cited and included in the reference list of the original manuscript.

11.   The interval of color bar is too large to indicate the anomalies between simulations and observation in Fig. 2 and Fig. 3. Please redraw the figures.

In accordance with the referees' wishes, we have now adjusted the colorbar of Figs. 2 (below) and 3 (see under comment #5).

[Figure]

**Figure 2: Absolute 2-m air temperature bias with respect to (a, d) the CRU observational dataset for (b, e) the UofT-CCSM4 and (c, f) the WRF ensemble in (a, b, c) JJAS and (d, e, f) DJF. The topography contours of 500, 1000, 2000, and 4000 m are also shown in black.**

12. The legend of "WRF1/2/3/4" in the figures might be replaced by WRF and the abbreviation of cumulus parameterization or specific name.

In accordance with the referees' wishes, we have now changed the legend of "WRF1/2/3/4" in the Figs. 3, 6 and 8-10 to "WRF-KF, WRF-GF, WRF-Tiedtke, WRF-BMJ".

13. Given that the line of "WRF ensemble" in the figures is overlayed by the lines of single experiments, it is hard to define the relation among different lines sometimes.

We changed the axis scale to fit the axes box more tightly around the data in Figs. 8-10. In other words, we tried
to zoom in as much as possible so that the readers can view the different lines more clearly. However, the lines of "WRF ensemble" are still sometimes overlayed by the lines of single experiments and this simply means their results are very similar in magnitude. In Figs. 11 and 12, we now use error bands to show the distribution range for all ensemble members instead of separate lines for each ensemble member.

[Figure]

**Figure 11: Monthly anomalies of (a-c) temperature and (d-f) total and solid precipitation over the three basins.**

[Figure]

**Figure 12: Monthly anomalies of (a-c) net precipitation and snowmelt and (d-f) total runoff from WRF over the three basins.**

**Response to the Comments of Referee 2 on the paper "Mid-Holocene climate of the Tibetan Plateau and hydroclimate in three major river basins based on high-resolution regional climate simulations" by Yiling Huo, W. R. Peltier and Deepak Chandan**

We thank the referee for his/her valuable comments on the content of our manuscript and his/her suggestions for improving the document. Following the reviewer's suggestions and comments, we have carefully revised our manuscript. We believe that the revised version satisfactorily addresses the referee's questions and concerns. In this reply, we respond to the issues, raised by the referee point by point. Our responses to the individual comments are shown in red text following the comments in black. For convenience, the modifications made to the text will also be shown in red.

Using WRF simulation, the authors tried to explore the changes to the river-headwater hydrological regimes on the TP during the mid-Holocene period. They found that dynamical downscaling enhances regional climate simulations over the TP in modern-day and MH climates and highlighted that they could overcome the cold biases, a typical issue across the Himalayas and TP region. The study demonstrated orbital factors' role in the seasonal precipitation cycle. Overall, the study is nice; there are some potentially fascinating points that they could have highlighted rather than simply summarizing the known MH climate.

Recommendation: Minor revision

Many thanks to the reviewer for this overall positive feedback, which we appreciate. We agree that the readers will likely find our paper new and interesting.

1)According to the authors, the ocean component was modified to make it more acceptable for paleoclimate simulations. Is this taken into account in MH and PI simulations? That would be nice to discuss it briefly if so.

Thank you for this suggestion. Lines 120-124 now state: "UofT-CCSM4 is based on the standard CCSM4 (Gent et al., 2011), but specific modifications have been made to the diapycnal diffusivity of the ocean component to make it more appropriate for paleoclimate simulations (Peltier and Vettoretti, 2014; Chandan and Peltier, 2017). These changes have been taken into account in both the MH and PI simulations to avoid introducing ambiguity related to different diapycnal mixing schemes when these two simulations are compared."

2) In Figures 5a & 5b, the authors attributed the changes to MH orbital and GHG forcings. So is this means the GHG forcing is different in MH and PI?

Yes. We now stated in lines 129-130 that "Compared to the PI, the $MH_{REF}$ experiments are forced by precessionally enhanced boreal summer insolation and slightly lower greenhouse-gas concentrations (Otto-Bliesner et al., 2017)."

3) If the GS only caused a 20% difference in precipitation, Is this coming from Saharan vegetation changes via ocean-atmosphere teleconnections? Is there a significant difference in SST forcing with and without GS? If so, it is better to include a brief description of this in the manuscript.

Yes. The presence of GS conditions in northern Africa shifts the Walker Circulation westward through changes in equatorial Atlantic SSTs and warms the Indian Ocean, which enhances the summer precipitation over Asia. Although a detailed discussion of the difference in SST forcing was presented in Huo et al. (2021) for the MH monsoons in South and Southeast Asia, we have chosen to highlight some of its conclusions here in accordance with the referees' wishes by adding one more figure and the following discussion illustrating the changes in SST

and atmospheric circulation in the global model in lines 279-286:

"The enhanced precipitation in MH experiment with Saharan vegetation is probably owing to ocean-atmosphere teleconnections as suggested in previous studies (Huo et al. 2021; Pausata et al., 2017). An albedo-induced warming develops over the vegetated Sahara, leading to a strong intensification and northward expansion of the West African Monsoon and a significant tropical North Atlantic SST warming (Fig. 7b; Pausata et al., 2016), which in turn changes atmospheric circulation and induces a notable intensification and westward extension of the Walker Circulation over the Pacific Ocean in the MH$_{GS}$ (Fig. 7d) compared to the MH$_{REF}$ experiment (Fig. 7c). The changes in the Walker Circulation weakens the low-level easterly winds over the eastern Pacific, but enhances easterly anomalies over the northern Indo-Pacific Ocean (Fig. 7b), which suppresses ENSO activity and enhances the Asian monsoon (Pausata et al., 2017)."

[Figure]

**Figure 7: SST (shaded, ° C), precipitation (shaded, mm d$^{-1}$) and 850 hPa winds (vector, m s$^{-1}$) anomalies during JJAS from the UofT-CCSM4 for (a) MH$_{REF}$ and (b) MH$_{GS}$. The topography contours of 500 m, 1000 m, 2000 m and 4000 m are also shown. PI climatological zonal stream function of the Walker circulation (contours: $0.2 \times 10^{11}$ kg s$^{-1}$ interval from $-2$ to $2 \times 10^{11}$ kg s$^{-1}$; 0 line in bold) and associated changes (shaded) in (c) MH$_{REF}$ and (d) MH$_{GS}$ relative to the PI.**

4) The river basin analysis is interesting. However, the authors did not give this section much weight in the abstract. This could have highlighted instead focusing on other well-known MH features. However, this section is too elaborate as well.

At the end of the abstract we now state:

"The simulation results were first validated over the upper basins of the three rivers before the hydrological
responses to the MH forcing for the three basins were quantified. Both the upper Yellow and Yangtze rivers exhibit a decline in streamflow during the MH, especially in summer, which is a combined effect of less snowmelt and stronger evapotranspiration. The GS forcing caused a rise in temperature during the MH, as well as larger rainfall but less snowfall and greater evaporative water losses. The Brahmaputra River runoff is simulated to increase in the MH, due to greater net precipitation."

We have also restructed section 4 and shortened it by around 1/3.

5) The authors noted the need for sufficient resolution to simulate TP on page 14, line 445. Is that, however, a huge deal in a model? even at coarse resolution, GCM is adequate to depict TP properly to a greater extent because this is a big area. Many researchers also mentioned how the Himalayas and TP play a minor role instead. How will the authors address these opposing issues? If the study does not shed light on this topic, it is preferable
to omit such extraneous descriptions rather than a casual sentence.

The model resolution may have a great impact on the simulation of the regional climate over the TP. In our manuscript, significant such discussion will be found regarding how our regional model with higher resolution outperforms the coarse-resolution global model in terms of temperature and precipitation simulation when compared to observations during the historical period and proxy data during the MH. Has the referee missed these?
Although we agree with the referee that some researchers argued that the South Asian summer monsoon circulation is unaffected by removal of the plateau, provided that the narrow orography of the Himalayas and adjacent mountain ranges is preserved, we are unaware of any studies that state both the TP and Himalayas are unimportant. Moreover, the focus of our study is the regional climate over the TP not South Asia. If there are specific references regarding the role of the Himalayas and TP on the TP regional climate that the referee believes
we have missed, it would have been more helpful to have listed them.

6) Again, the conclusion section also gave the least highlight to the quantifications over Riverhead regions.

In the conclusion, we rewrote the quantifications over the river basins in lines 436-444 and shortened the other parts:

"Both UYEB and UYAB hydrological regimes exhibit changes in the MH as manifested by a decline in
streamflow, especially in summer. Such flow decreases are a combined effect of changes in snowmelt and evapotranspiration. A significant amount of solid precipitation shifts to liquid precipitation and the JJAS net precipitation is simulated to shrink in both MH experiments. The GS forcing caused a rise in temperature during the MH, as well as larger rainfall but smaller snowmelt and larger evaporative water losses compared to the MH$_{REF}$. In the UBB, the simulated annual total precipitation increase in the MH is the largest among three basins, and, unlike the other two basins, the most significant MH hydroclimatic anomaly over the UBB may be an increase in runoff in both MH experiments, particularly in mid-summer, due to greater net precipitation. The greening of the Sahara led to higher temperature and enhanced snowmelt in spring and eliminated the drop in runoff in April in MH$_{REF}$."

7) The main point they suppose to express through the manuscript was land surface coupling and its importance.

But they have not taken care of this part properly in the manuscript. This could have been brought more interestingly in the conclusion part.

We have added a figure and some more discussion regarding the forcing due to Saharan vegetation (see under comment #3). In the conclusion part, we now added in lines 431-433:

"Saharan vegetation plays a crucial role in intensifying the West African Monsoon and modulating the atmospheric circulation, which alters the Walker circulation and increases Asian monsoon precipitation, through changes in equatorial Atlantic SSTs (Fig. 7)."

---

## Author Response (AR2)

**Response to the Comments of Referee 1 on EGUSPHERE-2022-40:**

**Title: Mid-Holocene climate of the Tibetan Plateau and hydroclimate in three major river basins based on high-resolution regional climate simulations**

**Authors: Yiling Huo, William Richard Peltier and Deepak Chandan**

5    We thank the referee for his/her valuable comments on the content of our manuscript and his/her suggestions for further improving the document. Following the reviewer's suggestions and comments, we have again carefully revised our manuscript. We believe that the revised version satisfactorily addresses the referee's questions and concerns. In this reply, we respond to the issues, raised by the referee point by point. Our responses to the individual comments are shown in red text following the comments in black.

10   The authors have addressed all comments and put a lot of effort into revising the manuscript. They well responded to all major issues and revised the manuscript as raised by the two Referees. Therefore, I recommend this manuscript for publication in the journal after minor revision with following shortcomings:

Minor comments

1. The map in Figs. 7a and b might mislead the distribution of longitude in Figs. 7c and d. Is it possible to unify

15   the distribution of longitude?

Thank you for this suggestion. We have unified the distribution of longitude and redrawn Fig 7. Now the longitude of all the panels starts and ends at 75◦ W.

[Figure]

**Figure 7: SST (shaded, ° C), precipitation (shaded, mm d⁻¹) and 850 hPa winds (vector, m s⁻¹) anomalies during JJAS from the UofT-CCSM4 for (a) MH$_{REF}$ and (b) MH$_{GS}$. The topography contours of 500 m, 1000 m, 2000 m and 4000 m are also shown. PI climatological zonal stream function of the Walker circulation (contours: 0.2 × 10¹¹ kg s⁻¹ interval from −2 to 2 × 10¹¹ kg s⁻¹; 0 line in bold) and associated changes (shaded) in (c) MH$_{REF}$ and (d) MH$_{GS}$ relative to the PI.**

2. Line 303: Do you mean Figs.8d and Figs. 9d?

We apologize for this error, and we have corrected the text.

25     3. Line 389: Please check the format of the units.

We apologize for this error, and we have changed the format of the units from mm d-1 to mm $d^{-1}$.